# ROBOTIC PROGRAMMER: VIDEO INSTRUCTED POLICY CODE GENERATION FOR ROBOTIC MANIPULATION

## ABSTRACT

Zero-shot generalization across various robots, tasks and environments remains a significant challenge in robotic manipulation. Policy code generation methods use executable code to connect high-level task descriptions and low-level action sequences, leveraging the generalization capabilities of large language models and atomic skill libraries. In this work, we propose **Robo**tic **Pro**grammer (**RoboPro**), a robotic foundation model, enabling the capability of perceiving visual information and following free-form instructions to perform robotic manipulation with policy code in a zero-shot manner. To address low efficiency and high cost in collecting runtime code data for robotic tasks, we devise Video2Code to synthesize executable code from extensive videos in-the-wild with off-the-shelf vision-language model and code-domain large language model. Extensive experiments show that RoboPro achieves the state-of-the-art zero-shot performance on robotic manipulation in both simulators and real-world environments. Specifically, the zero-shot success rate of RoboPro on RLBench surpasses the state-of-the-art model GPT-4o by 11.6%, which is even comparable to a strong supervised training baseline. Furthermore, RoboPro is robust to different robotic configurations, and demonstrates broad visual understanding in general VQA tasks.

## 1 INTRODUCTION

A long-term goal of embodied intelligence research is to develop a single model capable of solving any task defined by the user. Recent years have witnessed a trend towards large-scale foundation models on natural language processing tasks (Achiam et al., 2023; Touvron et al., 2023). Scaling up these language models in terms of model size and training tokens significantly improves the few-shot performance on a range of end tasks, even achieving performance comparable to previous state-of-the-art fine-tuning methods. However, for robotic tasks, we have yet to see large-scale pre-trained models that can directly transfer across different robots, tasks and environments without additional fine-tuning.

To improve the zero-shot generalization ability of robotic models, one common approach is to unify different tasks as the next action prediction. This paradigm requires the model to directly generate low-level action sequences. Brohan et al. (2023a); Padalkar et al. (2023); Kim et al. (2024); Niu et al. (2024) collected large amount of trajectories across various robots, tasks and environments. They trained vision-language-action (VLA) models derived from LLMs to map images and task instructions into discrete action tokens. Despite these models achieve better performance and show the capacity to transfer on novel objects and different tasks, fine-tuning is still required when deploying on new robots and environments. Besides, it is extremely expensive to collect trajectories through real-world robots, while using human-built simulators often leads to lack of diversity and introduces additional gap between simulation platform and real-world usages.

Another line of research aims to use code as compromise solution for bridging high-level instructions and low-level robot execution, leveraging the generalization capabilities of Large Language Models (LLMs) and atomic skill libraries. RoboCodeX (Mu et al., 2024) utilizes large vision-language model (VLM) to generate tree-of-thought plans and grasp preference. However, it also relies on manually-built simulation environment and human-annotated code for data curation, which is expensive and not friendly for scaling up in terms of training data.

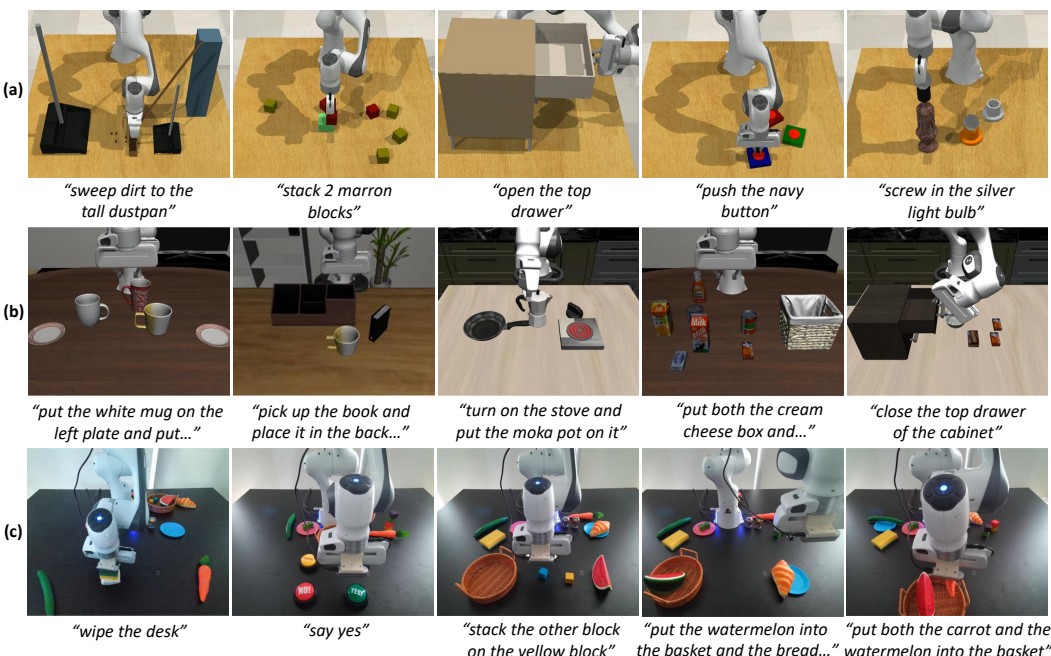

Figure 1: Visualization of evaluation tasks and execution results. RoboPro shows impressive zero-shot performance on novel and compositional tasks in RLBench (a), long-termed manipulation tasks in LIBERO (b), and real-world tasks (c). Video demos can be found in our supplementary materials.

In this work, we introduce **Robo**tic **Pro**grammer (**RoboPro**), a robotic foundation model, enabling the capability of perceiving visual information and following free-form user instructions to perform manipulation tasks without additional fine-tuning. RoboPro generates the executable code to connect high-level instructions and low-level action sequences. To address low efficiency and high cost in collecting runtime code data for robotic tasks, we devise Video2Code, an automatic data curation pipeline for multimodal code generation.

We draw our inspiration from the extensive amount of operational videos in-the-wild that implicitly contain necessary procedural knowledge about how to finish operational tasks. Previous research has focused on utilizing videos for large-scale supervised learning (Brohan et al., 2023a; Kim et al., 2024; Niu et al., 2024) or extracting relevant knowledge (e.g., affordance (Bahl et al., 2023)), while extracting executable policy code from videos is still under-explored. Our data curation pipeline uses the off-the-shelf VLM and Code LLM to synthesize code execution data from videos, which is much more efficient and scalable compared with generating code data from manually-built simulation environments. With Video2Code, we synthesize 115k robot execution code data along with the corresponding scene information and task descriptions from DROID (Khazatsky et al., 2024). Extensive experiments (examples depicted in Figure 1) show that RoboPro achieves the state-of-the-art zero-shot performance on robotic manipulation tasks in both simulators and real-world environments. Specifically, the zero-shot success rate of RoboPro on RLBench outperforms the state-of-the-art model GPT-4o by a gain of 11.6%. It is even comparable to a strong supervised training method PerAct (Shridhar et al., 2023). Furthermore, RoboPro is robust to different robotic configurations, and shows broad visual understanding on general VQA tasks.

## 2 RELATED WORKS

**Language-guided robot manipulation.** Language-conditioned robot manipulation refers to the use of natural language instructions to guide robotic actions. Natural language instructions allow non-experts to interact with robots through intuitive commands and enable robots to generalize to various tasks based on natural language input (Winograd, 1971). Recent advancements in language-conditioned embodied agents have leveraged Transformers (Vaswani et al., 2017) to enhance perfor-

mance on multi-task settings. One category of recent approaches is language-conditioned behavior cloning (BC), where models learn to mimic demonstrated language-conditioned actions and output dense action sequences directly. 3D BC methods (Shridhar et al., 2023; Zhang et al., 2024) trained from scratch perform well on specific environment, while lacking of generalization ability across environments. Vision-language-action (VLA) models (Brohan et al., 2023a; Kim et al., 2024; Niu et al., 2024) built on pre-trained large language models (LLMs) show capacity to transfer on novel objects and task settings, but need additional fine-tuning when being deployed on new environments and robots. Another line is to create high-level planners based on LLMs (Huang et al., 2022; Brohan et al., 2023b; Driess et al., 2023; Huang et al., 2023c), which output step-by-step natural language plans according to human instructions and environmental information. These methods show better generalization ability across environments, leveraging the reasoning and generalization ability of LLMs on language instructions and environments. However, there is still a gap between generated natural language plans and low-level robotic execution, requiring an extra step to score potential actions or decompose plans into relevant policies (Singh et al., 2023).

**Robot-centric policy code generation.** Code-as-Policies (Liang et al., 2023) proposes that executable code can serve as a more expressive way to bridge high-level task descriptions and low-level execution. Atomic skills to perceive 3D environments and plan primitive tasks are provided in predefined API libraries. LLMs process textual inputs and generate executable policy code conditioned on the API libraries (Liang et al., 2023; Huang et al., 2023a;b; Xu et al., 2023; Vemprala et al., 2024; Singh et al., 2023). RoboScript (Chen et al., 2024) further suggests that unified interface facilitates LLM's adaptability across different environments and hardware platforms. However, these methods rely solely on linguistic inputs, requiring detailed descriptions of environments and instructions as textual inputs, which limits their generalization and visual reasoning ability across environments. RoboCodeX (Mu et al., 2024) utilizes large vision-language model (VLM) to decompose multimodal information into object-centric units in a tree-of-thought format. Nevertheless, it relies on manually-built simulation environments and human-annotated data, which lacks environmental richness and is expensive for scaling up. Different from previous works using language-only LLMs, RoboPro enables visual reasoning ability and follows free-form instructions in a zero-shot manner. Furthermore, an automatic and scalable data curation pipeline Video2Code is developed to synthesize runtime code data from extensive videos in-the-wild in a quite efficient and low-cost fashion.

## 3 METHOD

### 3.1 PROBLEM STATEMENT

We consider language-guided robotic manipulation where each task is described with a free-form language instruction $I$. Given RGBD data from the wrist camera as the observation space $O_t$ and robot low-dimension state $s_t$ (e.g., gripper pose at current time $t$), the central problem investigated in this work is how to generate motion trajectories $T$, where $T$ denotes a sequence of end-effector waypoints to be executed by an Operational Space Controller (Khatib, 1987). However, generating dense motion trajectories at once according to the free-form instruction $I$ is quite challenging, as $I$ can be arbitrarily long-horizon and would require comprehensive contextual understanding. Policy code generation methods map long-horizon instructions to a diverse set of atomic skills, leading to rapid adaptation capabilities across various robotic platforms. With comprehensive contextual understanding and advanced visual grounding capabilities, large vision-language models can function as intelligent planners, translating the task execution process into generated programs due to their robust emergent capabilities.

To prompt vision-language models (VLMs) to generate policy code, we assume a set of parameterized skills with unified interface, which is defined as the API library $L_{\text{API}}$. $L_{\text{API}}$ can be categorized into perception module $L_{\text{per}}$ and control module $L_{\text{con}}$ based on the API's role in task execution process. $L_{\text{per}}$ is tasked with segmenting the task-relevant part point cloud $\Pi_I$ and predicting the physical property $\phi_I$ of relevant objects, while $L_{\text{con}}$ predicts the contact pose of the gripper and generates the motion trajectory $T$ based on the output of $L_{\text{per}}$ and the current robot state $s_t$:

$$
\begin{aligned}
L_{\text{API}} =& \{L_{\text{per}}, L_{\text{con}}\} \\
\{\Pi_I, \phi_I\} =& L_{\text{per}}(O_t, I) \\
T =& L_{\text{con}}(s_t, \{\Pi_I, \phi_I\}).
\end{aligned}
\tag{1}
$$

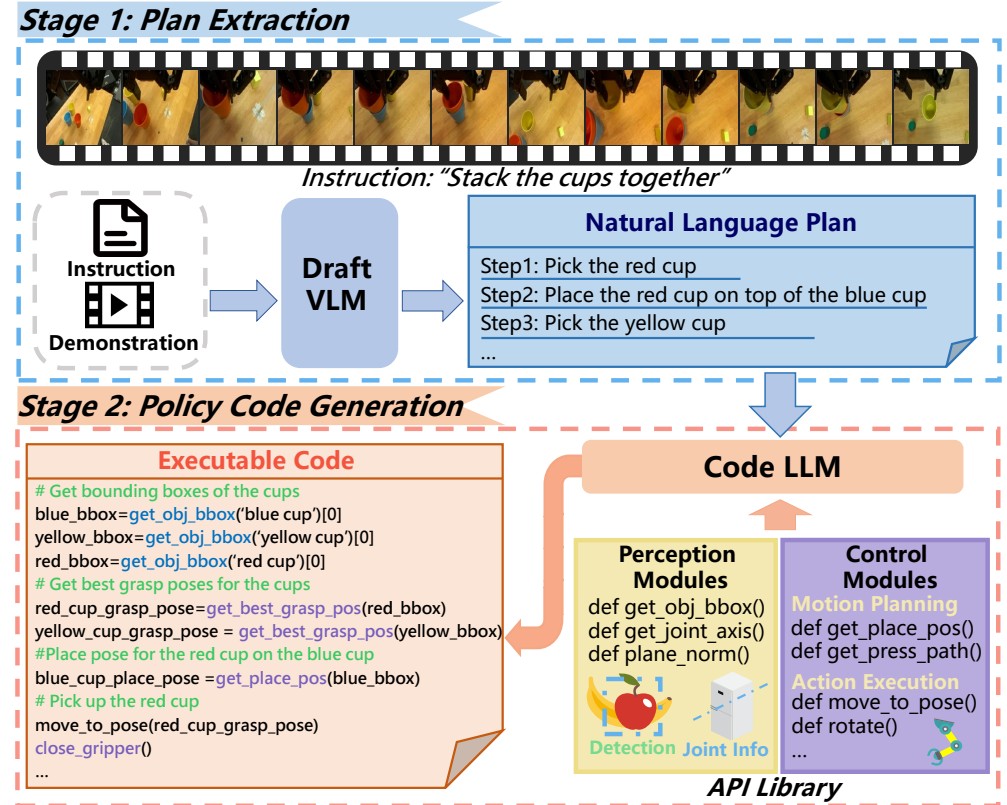

Figure 2: The data curation pipeline of Video2Code. We first use the Draft VLM to extract a brief natural language plan for execution of the user instruction. After that, the Code LLM generates robot-centric code using the provided API library and natural language plan from the first stage.

With the visual observation and the language instruction, VLMs generate executable policy code $\{\pi_i, p_i\}_{i=1}^N$ conditioned on the API library $L_{\text{API}}$, where $\pi_i$ denotes the $i$-th $L_{\text{per}}$ or $L_{\text{con}}$ calls and $p_i$ represents corresponding parameters for API calls. Each API call generates a sub-trajectory sequence $\tau_i$ of arbitrary length (the length is $\geq 0$). All sub-trajectory sequences $\{\tau_i\}_{i=1}^N$ are then concatenated to form the final complete motion trajectory $T$. The whole generation process is formulated as:

$$(O_t, I) \xRightarrow{\text{VLM}} \{\pi_i, p_i\}_{i=1}^N \Longrightarrow \{\tau_i\}_{i=1}^N. \tag{2}$$

Explainable API calls generated by VLMs connect the observation and high-level instructions to low-level execution, enabling the capacity of zero-shot generalization in free-form language instructions and across different environments. Obviously, training such VLMs to perceive environments, follow instructions and generate executable code will inevitably require a vast amount of diverse and well-aligned robot-centric multimodal runtime code data, which poses a significant challenge.

## 3.2 VIDEO2CODE: SYNTHESIZE ROBOTIC RUNTIME CODE FROM VIDEOS

Videos are widely available raw data sources for runtime code data synthesis. Extensive operational videos naturally provide low-level details of performing tasks such as *"how to pour tea into a cup"*, which inherently contain necessary procedural knowledge for runtime code data. Despite their favorable diversity and considerable quantity, it is still an under-explored and challenging problem how to collect executable policy code from demonstration videos efficiently. To this end, we devise Video2Code, a low-cost and automatic data curation pipeline to synthesize high-quality runtime code data from videos in an efficient way. Although open-source or lightweight vision-language models exhibit promising performance on video understanding tasks, a performance gap remains when compared to code-domain large language models in handling complex code generation tasks.

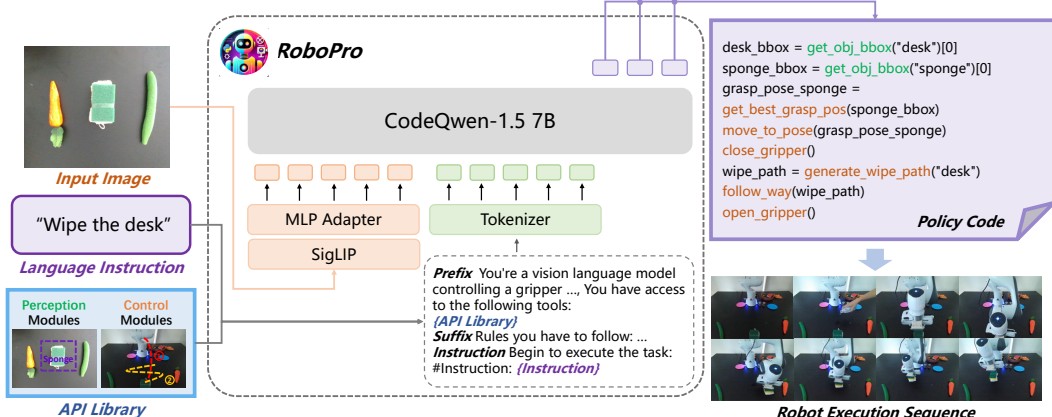

Figure 3: The overview of RoboPro. RoboPro utilizes environmental observation and natural language instruction as multimodal input, then outputs executable policy code. Extendable API library plays a role in mapping policy code into low-level execution sequences.

As depicted in Figure 2, to combine the visual reasoning ability of VLM and coding proficiency of code-domain LLM, Video2Code adopts a two-stage strategy.

**Plan extraction.** The first stage is to extract robot-centric plans in natural language from instructional videos. These instructional videos are filtered from DROID (Khazatsky et al., 2024), a large-scale robot manipulation dataset with 350 hours of interaction data across 564 scenes, 86 tasks, and 52 buildings. We extract 50k independent instructional videos with at least one free-form human instruction and further clip each video into 16 key frames. After that, we use Gemini-1.5-Flash (Team et al., 2023) as the Draft VLM to generate a brief list of actions for human instruction with these key frames as reference. As shown in Figure 2, the Draft VLM generates a step-by-step robot-centric plan from an instructional video to *"stack the cups together"*. The generated natural language plans contain knowledge and habit of human to follow free-form embodied instructions, and key visual information is extracted automatically from the instructional video.

**Policy code generation.** After plan extraction, we use Code LLM DeepSeek-Coder-V2 (Zhu et al., 2024) to "translate" these natural language plans into executable code. A complete prompt fed into the Code LLM includes API definitions, the natural language plan, and auxiliary part containing rules to follow. In the API definitions part, parameterized API functions are classified into two categories as formulated in Sec. 3.1: perception module, and control module. For each of these API functions, we provide API definitions and descriptions to demonstrate their usage. Auxiliary part contains prefix, third party tools, and rules to follow, similar to previous practices in RoboCodeX (Mu et al., 2024). Natural language plans accompanied with original human instructions are attached at the end of the prompt. As shown in Figure 2, step-by-step decomposed natural language plan guides the Code LLM to generate high-quality policy code in a Chain-of-Thought format. As for API implementation, we use GroundingDINO (Liu et al., 2023) and AnyGrasp (Fang et al., 2023) to get the bounding boxes and grasp preferences, respectively. Besides, we provide heuristic implementation for compositional skills. We finally collect 115k runtime code data with task descriptions and environmental observations using Video2Code for supervised fine-tuning.

### 3.3 ROBOPRO: ROBOTIC FOUNDATION MODEL

**Model architecture.** As shown in Figure 3, RoboPro has a vision encoder and a pre-trained LLM. They are connected with a lightweight adaptor layer consisting of a two-layer MLP. Specifically, the vision backbone first encodes the image into a sequence of visual tokens. After that, the lightweight adaptor is designed to project visual tokens onto embedding space of the LLM. In addition, we provide the API definitions and the user instruction as the text inputs. The visual and text tokens are directly concatenated and then fed into the LLM, as similarly done in Liu et al. (2024b). The LLM are trained to generate the runtime code based on the visual inputs and task description.

RoboPro is designed to reason on multimodal inputs and generate executable policy code for robotic manipulation. Thus, two key factors for the choice of its components are the ability of visual reasoning and the quality of policy code generation. RoboPro adopts SigLIP-L (Zhai et al., 2023) as the vision encoder, which yields favorable performance on general visual reasoning tasks. For the base LLM, a code-domain LLM, CodeQwen-1.5 (Bai et al., 2023), is utilized, which shows state-of-the-art performance among open-source code models. The model architecture and working process of RoboPro are illustrated in Figure 3.

**Training.** The training procedure of RoboPro consists of three stages: visual alignment, pre-training, and supervised fine-tuning (SFT). We first train a lightweight adaptor layer while freezing the vision encoder and LLM with LLaVA-Pretrain (Liu et al., 2024b). Then we pre-train the lightweight adaptor and the LLM on a corpus of high-quality image-text pairs (Chen et al., 2023). For supervised fine-tuning, the 115k runtime code data generated by Video2Code (as noted in Sec. 3.2) are used. To avoid overfitting and enhance visual reasoning ability, a general vision language fine-tuning dataset (LLaVA-1.5 (Liu et al., 2024b)) is also involved during the SFT process. Thus, RoboPro is trained to follow free-form language instructions and perceive visual information to generate executable policy code for robotic manipulation. Meanwhile, it exhibits broad visual understanding to perform general VQA tasks. Our code and model will be released to the public.

# 4 EXPERIMENTS

## 4.1 ZERO-SHOT ROBOTIC MANIPULATION

**Setup.** Following PerAct (Shridhar et al., 2023), we select 9 tasks with the requirement of novel instruction understanding or long-horizon reasoning in RLBench (James et al., 2020) for evaluation. Each task is evaluated with 25 episodes scored either 0 or 100 for failure or success in task execution. Detailed experiment settings and task information in RLBench can be found in Appendix A.1.

**Baselines.** The baselines can be categorized into two groups. One common approach requires supervised training on the simulation platform, e.g., behavior cloning methods, including PerAct (Shridhar et al., 2023) and LLARVA (Niu et al., 2024). They are either trained from scratch or fine-tuned with hundreds of episodes from RLBench. PerAct is trained on 100 episodes, and LLARVA is fine-tuned on 800 episodes per task in RLBench. The methods from another group do not require additional training. They first output robot-centric policy code, then execute it with provided APIs. We evaluate their zero-shot performance on RLBench. CaP (Liang et al., 2023) equips large language model with the ground-truth textual scene descriptions, containing object names, attributes, and instructions, to generate executable code. Following their paper, we implement CaP with GPT-3.5-Turbo (gpt-3.5-turbo-0125). GPT-4o (OpenAI (2024), gpt-4o-2024-05-13) is the state-of-the-art multimodal model for various vision-language tasks. For RoboPro and GPT-4o, we require the model to directly generate the executable code given the image from the wrist camera, user instructions and API definitions. We also analyze the generalization ability of RoboPro on the formation of API libraries, which is further elaborated in Sec. 4.2. For a fair comparison, we adopt the same API library for these methods (i.e., CaP, GPT-4o, and RoboPro). Our API library shares similar design formulation as RoboCodeX (Mu et al., 2024)[1], with detailed implementation in Appendix B.

**Results.** We report the average success rate on 25 episodes for each task. As shown in Table 1, the zero-shot result of RoboPro surpasses language-only policy code generation method (CaP) by 19.1%. Besides, our model significantly outperforms the state-of-the-art VLM GPT-4o by 11.6% on average success rate. More importantly, the zero-shot success rate of Robo-Pro is even comparable with a strong behavior-cloning baseline PerAct that requires supervised training. It

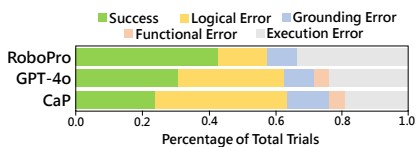

Figure 4: Error breakdown on RLBench.

demonstrates the effectiveness of our model for manipulation tasks. To thoroughly analyze the factors

---

[1]RoboCodeX was not compared in our experiments as this work has not released its model checkpoints and its original evaluations were not conducted on publicly available simulation platforms.

Table 1: Success rate (%) on RLBench Multi-Task setting. Methods greyed on need supervised training on the simulation platform.

| Models | Push Buttons | Stack Blocks | Open Drawer | Close Jar | Stack Cups | Sweep Dirt | Slide Block | Screw Bulb | Put in Board | Avg. |
|---|---|---|---|---|---|---|---|---|---|---|
| *Specialists, w/ training on RLBench* | | | | | | | | | | |
| LLARVA (Niu et al., 2024) | 56 | 0 | 60 | 28 | 0 | 84 | 100 | 8 | 0 | 37.3 |
| PerAct (Shridhar et al., 2023) | 48 | 36 | 80 | 60 | 0 | 56 | 72 | 24 | 16 | 43.6 |
| *Generalists, w/o training on RLBench* | | | | | | | | | | |
| CaP (Liang et al., 2023) | **72** | 4 | 24 | 40 | 0 | 36 | 4 | 20 | **12** | 23.6 |
| GPT-4o (OpenAI, 2024) | **72** | 20 | 56 | 36 | 4 | 40 | 20 | 20 | **12** | 31.1 |
| **RoboPro (ours)** | 68 | **48** | 68 | 44 | 4 | **48** | 60 | 32 | **12** | **42.7** |
| w/ API Renaming | 68 | 40 | 60 | **48** | 4 | **48** | 68 | **36** | **12** | **42.7** |
| w/ API Refactoring | 68 | 36 | **72** | 44 | **8** | 16 | **80** | 28 | **12** | 40.4 |

Table 2: Success rate (%) on 8 tasks in LIBERO. The result of PerAct is a zero-shot transfer from RLBench to LIBERO.

| Models | Turn on Stove | Close Cabinet | Put in Sauce | Put in Butter | Put in Cheese | Place Book | Boil Water | Identify Plate | Avg. |
|---|---|---|---|---|---|---|---|---|---|
| *Specialists, RLBench → LIBERO* | | | | | | | | | |
| PerAct (Shridhar et al., 2023) | 0 | 0 | 0 | 0 | 0 | 0 | 0 | 0 | 0 |
| *Generalists* | | | | | | | | | |
| CaP (Liang et al., 2023) | 0 | 37 | 17 | 13 | 7 | 30 | 7 | 7 | 14.8 |
| GPT-4o (OpenAI, 2024) | 37 | 17 | 63 | 43 | 57 | **43** | 17 | 3 | 35.0 |
| **RoboPro (ours)** | **97** | **60** | **67** | **53** | **63** | **43** | **23** | **13** | **52.4** |

contributing to the performance gap between different methods, we conducted an error breakdown for the policy code generation approaches. In the context of policy code generation methods, the successful execution of manipulation tasks relies on both the accuracy of the policy code and the capabilities of the API library. The main types of errors impacting the quality of robot-centric policy code are logical errors, functional errors, and grounding errors. These errors are associated with challenges in the appropriate selection and utility of APIs, as well as issues related to visual grounding. As depicted in Figure 4, the results show that all these methods perform well on following functional definition of API library, causing a low occupancy of functional error. Compared with linguistic only method CaP, GPT-4o and RoboPro show a noticeable improvement in target object grounding. The main failure cases of CaP and GPT-4o fall in logical error, including API selection and proper order of API calls. In contrast, RoboPro effectively reduces this margin, mainly owing to the procedural knowledge about long-term execution learned in Video2Code. Execution errors maintain a consistent proportional relationship with successful cases, which result from API limitations rather than inaccuracies in the policy code. Detailed illustration of error cases can be found at Appendix A.4.

## 4.2 ZERO-SHOT GENERALIZATION

In supervised training methods, outstanding performance is often achieved in familiar environments. However, due to data scarcity, challenges arise in terms of generalizing across different environments and robot configurations in a zero-shot manner. Different environments mainly introduce variations in tasks, context, and objects, while robot configurations refer to changes of degrees of freedom, action spaces in different robotic embodiments. For policy code generation methods, different robot configurations are mainly reflected in variations in the formats of APIs (e.g. input-output format). Additionally, users may have their own preferences when customizing API libraries for similar functions. We believe that a robust policy code generation model should also demonstrate strong

adaptability to these variations. To evaluate the zero-shot generalization ability of RoboPro, we conduct experiments in two aspects: across different environments and across different API libraries.

**Generalization across different environments.** We use LIBERO (Liu et al., 2024a) as an extra simulator to evaluate the zero-shot generalization across different environments. We choose 8 representative tasks from LIBERO-100 as the evaluation set. Each task is evaluated with 30 episodes. These tasks include short-horizon tasks which need scene understanding, and long-horizon tasks which require multi-step implementation. Detailed task descriptions and corresponding examples can be found in Appendix A.2. For behavior cloning method PerAct, we evaluate its model trained on RLBench as described in Sec. 4.1, which will be tested on LIBERO without further fine-tuning. For CaP, GPT-4o and RoboPro, we evaluate their zero-shot performance. As reported in Table 2, PerAct trained on RLBench struggles on the tasks from LIBERO. It indicates that PerAct is difficult to generalize across different environments without additional fine-tuning. Furthermore, RoboPro significantly outperforms GPT-4o by a gain of 17.4% average success rate on 8 LIBERO tasks, which is aligned with the observations from the experiments on RLBench. Compared with GPT-4o, RoboPro executes more accurate sequences of actions to complete various manipulation tasks. For instance, when given the task *"Turn on the stove"*, RoboPro consistently approaches the stove knob, grasps it, and rotates it clockwise. In contrast, GPT-4o sometimes misinterprets the knob's affordance, attempting to press it rather than rotate.

**Generalization across different API libraries.** The formation and definition of pre-defined API library is a key factor that affects the performance of general robotic models, since they are usually deployed across different types of robots. Robustness to the changes of API library implies that the model can understand and internalize the atomic skills under the API interface. To assess the generalization of RoboPro under different level of changes in API library, we designed two representative sets of experiments: the *API Renaming* set and the *API Refactoring* set. For renamed APIs, we only change in their names and keep consistent in functional structure (e.g., the type of return values and arguments). For refactored APIs, we change in functional structure but keep their names. Take the control API `"get_best_grasp_pose()"` as an example. In the API Renaming set, it is renamed as `"generate_obj_grasp_pos()"` without changes on functionality, and in the API Refactoring set, the inputs, outputs and comments are all changed (e.g., the input format changes from `"bbox"` to `"np.ndarray"`). As shown in Table 1, the performance of RoboPro on RLBench is robust to the changes in API formation. The detailed implementations of renamed and refactored APIs can be found in Appendix B.

## 4.3 REAL-WORLD EXPERIMENTS

To evaluate the performance of RoboPro in real-world scenarios, we conduct realistic experiments on a Franka Emika robot arm equipped with an Intel RealSense D435i wrist camera. As emphasized in Sec. 3.1, long-horizon task decomposition and visual understanding capabilities are crucial for zero-shot generalization in language-guided robotic manipulation. To assess Robo-Pro's performance in these aspects, we carefully designed 8 tasks, ranging from short-horizon to long-horizon tasks, as well as tasks that require visual comprehension. For instance, RoboPro is required to select object with "wipe" affordance from the scene given instruction *"wipe the desk"*. Additionally, to rigorously validate RoboPro's generalization capability across different real-world scenarios, we ensure that each task

Table 3: The zero-shot success rate of Robo-Pro across 8 real-world manipulation tasks.

| Task | # Var | # Test | Succ. % |
|------|-------|--------|---------|
| Move in Direction | 2 | 10 | 80 |
| Setup Food | 2 | 10 | 90 |
| Distinct Base | 2 | 10 | 70 |
| Prepare Meal | 2 | 10 | 60 |
| Tidy Table | 2 | 10 | 70 |
| Express Words | 4 | 10 | 60 |
| Stack on Color | 5 | 10 | 50 |
| Wipe Desk | 2 | 10 | 100 |

consists of at least two variations (denoted as "# Var") in terms of object categories and physical properties (10 tests are run for each task). As shown in Table 3, RoboPro is able to achieve 72.5% success rate on average among all 8 tasks, which verifies RoboPro's strong generalization ability in real-world scenarios without any specific fine-tuning. We also observe RoboPro exhibits impressive emergent ability in visual reasoning. For example, as depicted in Figure 5, when asked to wipe the

Figure 5: Illustration of execution on visual reasoning tasks in real-world environment. RoboPro presses buttons to express words (a), stacks object in an appropriate order based on visual properties (b), and chooses the best tool to wipe the desk (c).

desk, RoboPro will choose the appropriate tool (the sponge) among irrelevant objects, and grasp it to wipe water on the desk. We also provide detailed real-world setup in Appendix A.3.

## 4.4 EVALUATION ON GENERAL VQA TASKS

As mentioned in Sec. 3.3, RoboPro can meanwhile exhibit broad visual understanding to perform general visual question answering. To evaluate this ability, we conduct experiments across a range of general VQA tasks. We compare RoboPro with InstructBLIP based on Vicuna 7B v1.1 (Dai et al., 2023), LLaVA-1.5 (Liu et al., 2024b) and ShareGPT4V (Chen et al., 2023). We evaluate the zero-shot performance of these models on perception and reasoning tasks with VQAv2 (Goyal et al., 2017), GQA (Hudson & Manning, 2019) and TextVQA (Singh et al., 2019). As shown in Table 4, RoboPro can not only generate executable code for

Table 4: The zero-shot accuracy of RoboPro and the baselines on general VQA tasks.

| Models | $VQA^{v2}$ | GQA | $VQA^T$ |
|---|---|---|---|
| InstructBLIP | - | 49.2 | 50.1 |
| LLaVA-1.5 | 78.5 | 62.0 | 58.2 |
| ShareGPT4V | 80.6 | 63.3 | 60.4 |
| **RoboPro** | **80.9** | **63.9** | **62.9** |

robotic control, but perform well on multimodal perception and reasoning tasks. The results indicate that our model demonstrates quite competitive performance on general VQA tasks compared to ShareGPT4V, which is the state-of-the-art vision-language model with similar model size. Experiments on general VQA tasks further confirm RoboPro's capability of comprehensive visual understanding, which is a key factor in its success of manipulation tasks.

## 4.5 ABLATION STUDY

We conduct extensive ablations to evaluate the effectiveness of Video2Code and the contributions of individual components in RoboPro and Video2Code framework. Specifically, we conduct ablation studies on the base LLM in RoboPro, as well as the Draft VLM and Code LLM in Video2Code. We provide detailed ablation results in Appendix A.5.

**Effectiveness of Video2Code.** We compare our model trained with and without Video2Code on manipulation and general VQA tasks. For a fair comparison, we only remove Video2Code from the fine-tuning stage for the baseline, that is, the 115k runtime code data are excluded and only the general vision language fine-tuning dataset is used during the SFT process, as described in Sec. 3.3. The first two rows of Table 5 show the comparison of the two settings. It is found that the Video2Code generated data have significantly improved the performance on both RLBench and LIBERO by a gain of 42.3% and 45.4%, respectively, which indicate Video2Code's efficacy in enhancing the ability of skills utility and instruction following. Moreover, our model trained with such code data can also bring slight improvement on general VQA tasks.

Table 5: Ablations of Video2Code and different base LLMs on manipulation and general VQA tasks.

| LLM | Video2Code | Manipulation | | General VQA | | |
|---|---|---|---|---|---|---|
| | | RLBench | LIBERO | $VQA^{v2}$ | GQA | $VQA^T$ |
| CodeQwen-1.5-7B | ✗ | 0.4 | 7.0 | 80.5 | 63.8 | 62.1 |
| CodeQwen-1.5-7B | ✓ | **42.7** | **52.4** | **80.9** | **63.9** | **62.9** |
| DeepSeek-Coder-6.7B | ✓ | 41.3 | 48.8 | 78.3 | 60.9 | 59.5 |

Table 6: The selection of the Draft VLM and Code LLM for Video2Code.

| Method | RLBench | LIBERO |
|---|---|---|
| MiniCPM-V + Gemini-1.5-Flash | 8.0 | 21.7 |
| Gemini-1.5-Flash + Gemini-1.5-Flash | 22.7 | 31.7 |
| **Gemini-1.5-Flash + DeepSeek-Coder-V2** | **42.7** | **52.4** |

**Choice of base LLM.** We further compare the performance of RoboPro using different code-domain base LLMs. Specifically, we choose DeepSeek-Coder-6.7B-Instruct (Guo et al., 2024) and CodeQwen-1.5-7B-Chat (Bai et al., 2023) for comparison. As shown in Table 5, RoboPro trained on CodeQwen-1.5-7B-Chat outperforms the version trained on DeepSeeK-Coder-6.7B-Instruct on both manipulation and general VQA tasks. These results demonstrate that employing a more powerful base LLM for code generation task can consequently enhance performance in both tasks.

**Choice of Draft VLM and Code LLM.** The Draft VLM and Code LLM are key components in the design of Video2Code. As stated in Sec. 3.2, we choose Gemini-1.5-Flash as Draft VLM and DeepSeek-Coder-V2 as Code LLM for default configurations. To analyze how the choice of Draft VLM and Code LLM effects the quality of runtime code data, we set three different combinations of Draft VLM and Code LLM for data curation. We choose Gemini-1.5-Flash and a light-weight VLM MiniCPM-V (Yao et al., 2024) for Draft VLM evaluation, while selecting a code domain LLM DeepSeek-Coder-V2 and a general VLM Gemini-1.5-Flash for Code LLM evaluation. All other settings are consistent with those in our main experiment. As shown in Table 6, enhanced visual reasoning capabilities of the Draft VLM, along with stronger code synthesis abilities of the Code LLM, both play a crucial role in curating high-quality runtime code data.

## 5 CONCLUSION AND FUTURE WORK

In this work, we propose RoboPro, a robotic foundation model, which perceives visual information and follows free-form instructions to perform robotic manipulation in a zero-shot manner. To address low efficiency and high cost for runtime code data synthesis, we propose Video2Code, a scalable and automatic data curation pipeline. Through extensive experiments, with assistance of Video2Code, RoboPro achieves impressive generalization capability compared with training-based methods, and exhibits significant improvement on performance compared with other policy code generation methods. These results indicate that incorporating procedural knowledge within operational videos into training process will bring substantially enhanced understanding of skills (i.e., API libraries) and free-form instructions. Beyond the scope of robotic manipulation tasks, policy code generation methods also show potential in many other robotic applications (e.g., navigation). In the future, we would like to expand our method to more application scenarios to provide more comprehensive support for complex real-world robotic deployments.

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

# A  TASK DETAILS

## A.1  TASKS IN RLBENCH

RLBench is a simulation platform set in CoppelaSim (Rohmer et al., 2013) and interfaced through PyRep (James et al., 2019). Robotic models control a 7-dof Franka Panda robot with a parallel gripper to complete language-conditioned tasks. RoboPro is evaluated on 9 tasks from RLBench (James et al., 2020). Modification on these tasks is consistent with PerAct (Shridhar et al., 2023). Each task in RLBench is provided with several variations on language instructions describing the goal. In order to validate RoboPro's adaptation ability across various and vague instructions, we pop out an instruction from the language template list for each episode during evaluation instead of just using the first language template. Detailed descriptions and modification for each task in RLBench are provided below.

**Push Buttons.**    Push down colored buttons in a specific order. The task has 20 different variances on the color of buttons, and three variances on the number of buttons to be manipulated. The success metric of this task is to push down specific buttons in correct order.

**Close Jar.**    Put the lid on the table onto the jar with specific color. This task also has 20 different variations on the color of the jars. The success metric is that the lid is on the top of the target jar, and the gripper doesn't grasp anything.

**Stack Blocks.**    Stack two to four blocks with specific color onto the green target area. There are always two groups of four blocks with the same color, and this task has 20 variations on the color of the blocks. The success metric has a further requirement that all stacked blocks inside the area of a green platform beyond the original language instruction. We add target prompt to specify the stacking area.

**Open Drawer.**    Open specific drawer of a cabinet. there are three different variations on the position of the drawer: top, middle, and bottom. The success metric is a full extension of the target drawer joint. Before execution, we first adjust the gripper position to face the cabinet.

**Stack Cups.**    Stack other two cups onto the cup with specific color. This task has 20 variations on the color of the cups. The success metric of this task is that the other cups are inside the target cup.

**Sweep Dirt.**    Sweep dirt particles to the target dustpan. There are two dustpans specified as a tall dustpan and a short dustpan. The success metric of this task is that all 5 dirt particles are in the target dustpan. This task is modified by PerAct.

**Slide Block.**    Slide the red cube in the scene to the target colored area. There are four areas with different color on each corner of the scene, and the cube cannot be picked up. The success metric is that the cube is inside the area with the target color, which is modified by PerAct.

**Screw Bulb.**    Screw light bulb with the specified base onto the lamp base. There are two bulbs in the scene at once, and the color of the holders have 20 different variations. The success metric is that the bulb is inside the lamp stand.

**Put in Board.**    Pick up the specified object and place it into the cupboard above. There are always 9 different objects on the table. The success rate is that the target object is in the cupboard.

A.2    TASKS IN LIBERO

In this section, we provide a detailed description of 8 tasks selected from the LIBERO-100 dataset. Each task is associated with a specific language instruction, with the task ID and corresponding instruction shown in Table 7. The tasks *"Turn on Stove"* and *"Close Cabinet"* are taken from LIBERO-90, which focuses on testing atomic skills and environmental understanding. The remaining tasks are more complex, requiring multi-step execution, and are selected from LIBERO-10. These 8 tasks challenge RoboPro to comprehend diverse visual environments and follow extended language instructions. As illustrated in Figure 6, the tasks encompass a wide range of robotic capabilities, including object selection, spatial reasoning, scene comprehension, and long-term execution.

Table 7: The manipulation tasks selected for the evaluation of zero-shot generalization on LIBERO.

| Task ID | Task Instruction |
| --- | --- |
| Turn on Stove | turn on the stove |
| Close Cabinet | close the top drawer of the cabinet |
| Put in Sauce | put both the alphabet soup and the tomato sauce in the basket |
| Put in Butter | put both the cream cheese box and the butter in the basket |
| Put in Cheese | put both the alphabet soup and the cream cheese box in the basket |
| Place Book | pick up the book and place it in the back compartment of the caddy |
| Boil Water | turn on the stove and put the moka pot on it |
| Identify Plate | put the white mug on the left plate and put the yellow and white mug on the right plate |

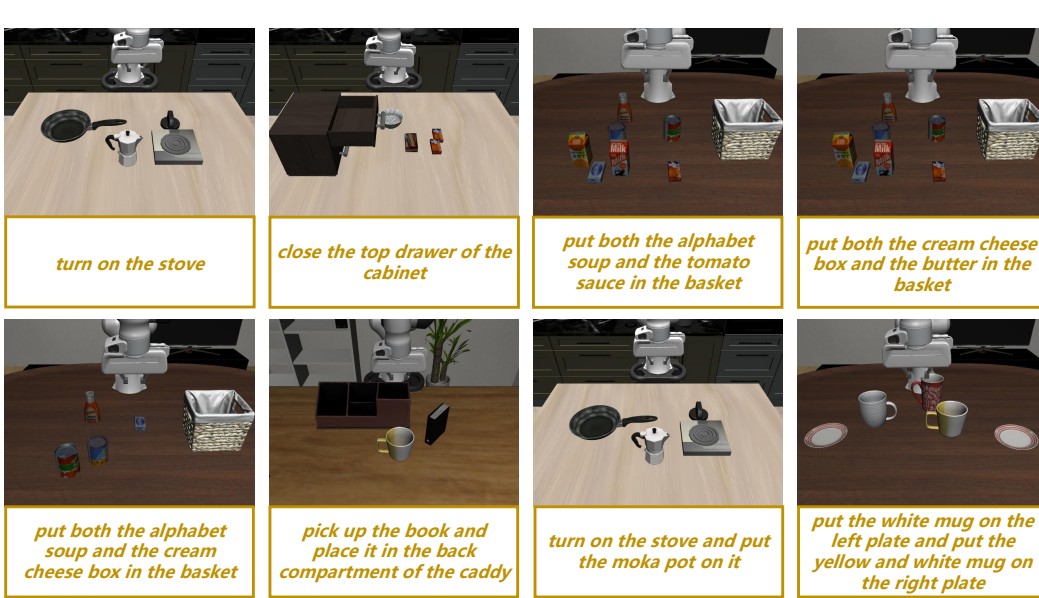

Figure 6: Illustration of the selected tasks from LIBERO benchmark.

### A.3 TASKS IN REAL-WORLD EXPERIMENTS

To validate the performance of RoboPro, the real-world experiments are implemented on a Franka Emika Panda robotic arm with a parallel jaw gripper, as shown in Figure 7. We use an Intel RealSense D435i camera to provide RGB-D input signals under the camera-in-hand setting. Easy-handeye ROS package is used to calibrate the extrinsics of the camera frame with respect to the robot base frame. For robot control, we use the open-source frankapy package to send real-time position-control commands to robot after receiving the control signals from RoboPro. During test time for each task, natural language instructions, extrinsic matrix, intrinsic matrix, current environment observation in the form of RGB-D image,

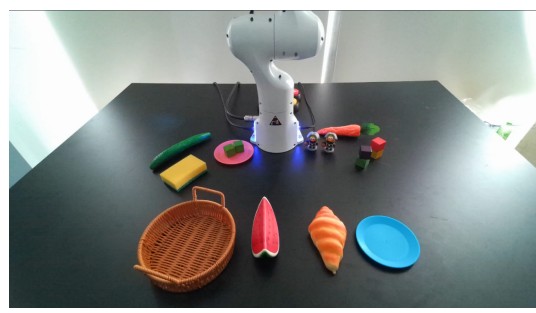

Figure 7: The setup for real-world experiments.

and the low dimensional state of the robot are prepared for RoboPro to generate corresponding 6-DOF action trajectories. Examples of all 8 real-world tasks with natural language instructions are illustrated in Figure 8.

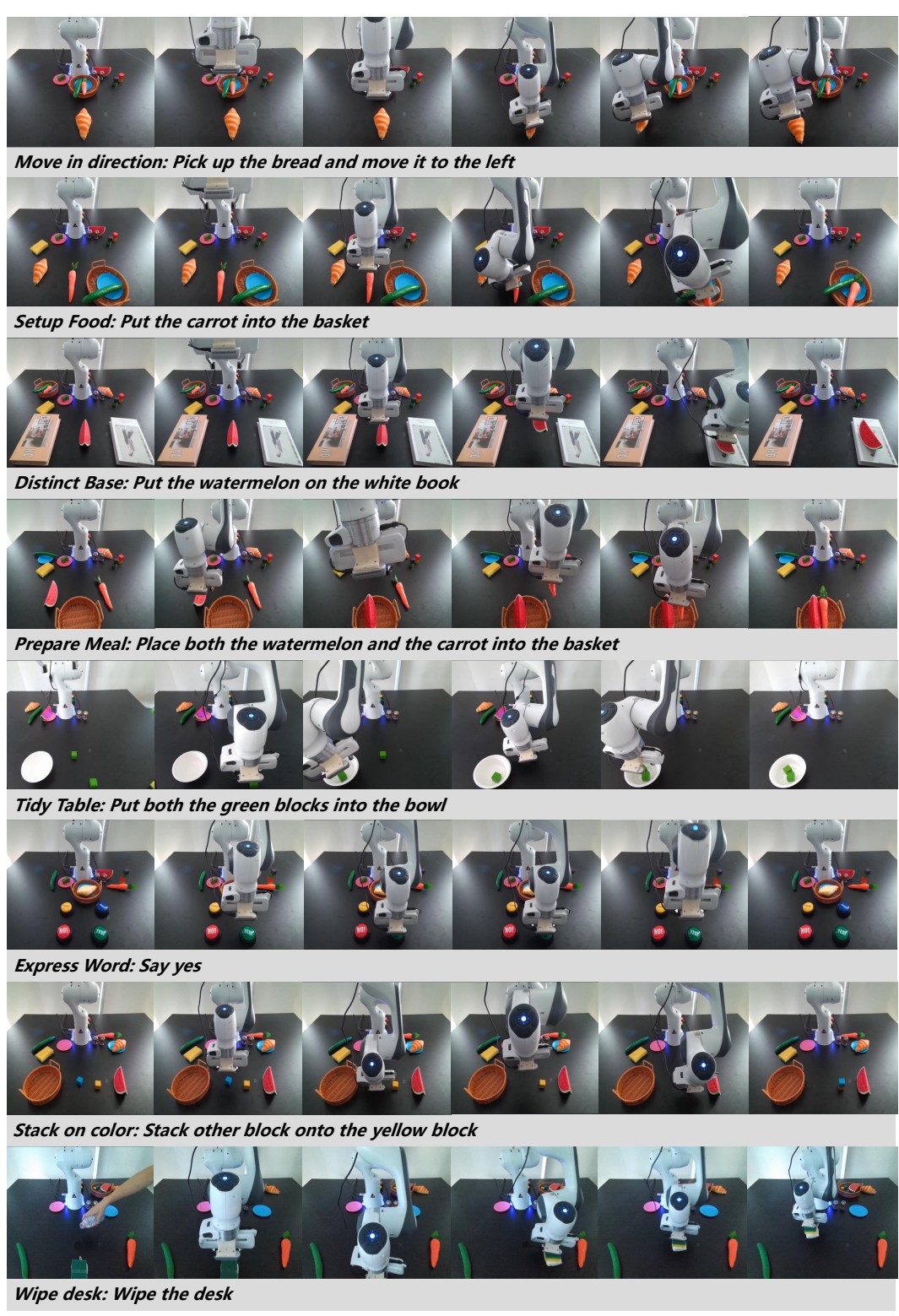

Figure 8: Illustration of RoboPro on the real-world experiments.

## A.4 ILLUSTRATION OF ERROR CASES

Figure 9: Illustration of failure and success cases of different policy code generation methods in LIBERO and RLBench.

## A.5 ADDITIONAL EXPERIMENT RESULTS

Table 8: Detailed success rate (%) of ablation study on the RLBench tasks.

| LLM | Video2Code | Push Buttons | Stack Blocks | Open Drawer | Close Jar | Stack Cups | Sweep Dirt | Slide Block | Screw Bulb | Put in Board | Avg. |
|---|---|---|---|---|---|---|---|---|---|---|---|
| CodeQwen-1.5-7B | ✗ | 0 | 0 | 0 | 0 | 0 | 0 | 0 | 4 | 0 | 0.4 |
| CodeQwen-1.5-7B | ✓ | 68 | **48** | 68 | 44 | **4** | 48 | 60 | **32** | 12 | **42.7** |
| DeepSeek-Coder-6.7B | ✓ | **72** | 32 | 68 | **48** | 0 | 24 | **84** | **32** | 12 | 41.3 |

Table 9: Detailed success rate (%) of ablation study on the LIBERO tasks.

| LLM | Video2Code | Turn on Stove | Close Cabinet | Put in Sauce | Put in Butter | Put in Cheese | Place Book | Boil Water | Identify Plate | Avg. |
|---|---|---|---|---|---|---|---|---|---|---|
| CodeQwen-1.5-7B | ✗ | 0 | 43 | 0 | 0 | 13 | 0 | 0 | 0 | 7.0 |
| CodeQwen-1.5-7B | ✓ | **97** | **60** | **67** | **53** | **63** | 43 | **23** | 13 | **52.4** |
| DeepSeek-Coder-6.7B | ✓ | **97** | **60** | 47 | **53** | 60 | **53** | 0 | **20** | 48.8 |

## B PROMPT AND API IMPLEMENTATIONS

Listing 1: An example of a full prompt in `RoboPro`

```
"""You're a vision language model controlling a gripper to complete manipulation tasks.
    Combine the images you see with the text instructions to generate detailed and workable
    code for the current scene.
You have access to the following tools:
"""
------------
import numpy as np
import torch
import math

#Perception Modules
def get_obj_bbox(description: str)->list[bbox]:
"""get the 2D boundingbox of all objects match description. When it comes to the specific
    parts or orientation of objects, the description should be detailed. Like 'handle of
    microwave', 'left side of shelf'.
Return: list[bbox: np.ndarray]"""

def get_best_grasp_pos(grasp_bbox: bbox):
"""get best grasp pose to grasp specific object.
Return: grasp_pose: Pose"""

def get_place_pos(holder_bbox: bbox):
"""Predict the place pose for an object relative to a holder
Args: holder_bbox: bbox of target region of the holder.
Return: place_pose: Pose"""

def get_joint_axis(joint_object_name: str):
"""Get the joint direction of an object
Args: joint_object_name: the name of object have joint axis.
Return: joint_axis: np.ndarray"""

def generate_joint_path(joint_axis: np.ndarray, open: bool):
"""Generate a gripper path of poses around the joint. open is True when need open container
    around joint, False when close container.
Return: path: list[Pose]
"""

def generate_slide_path(target: Optional[str] = None, direction: Optional[np.ndarray] = None):

"""Generate path of poses to slide or push object to target or in specific direction.
Args:
target: The target location. If provided, 'direction' must be None.
direction: The direction vector to slide the object along. If provided, 'target' must be None.

Return: path: list[Pose]
"""

def generate_sweep_path(object: Optional[str] = None, target: Optional[str] = None, direction:
        Optional[np.ndarray] = None):
"""This function is designed to generate movement paths for sweeping actions using tools such
    as sweepers, brooms. Grasp the tool before sweeping.
Args:
object: The object to be swept. If set to None, the function will perform a general sweeping.
target: The target area or location to sweep towards. If provided, 'direction' must be None.
direction: The direction vector for the sweeping motion. If provided, 'target' must be None.
Return: path: list[Pose]
```

```python
    """

def generate_wipe_path(region: str):
    """This function is designed to generate movement paths for wiping actions using tools such
        as towel, sponge. Grasp the tool before wiping.
    Args:
    region (str): region to be wiped or cleaned.
    Return: path: list[Pose]
    """

def generate_pour_path(grasped object: str, target: str):
    """Generate gripper path of poses to pour liquid in grasped object to target.
    Return: path: list[Pose]
    """

def generate_press_pose(bbox):
    """Get best pose to press or push buttons."""

#Action Modules
def move_to_pose(Pose):
    """Move the gripper to pose."""

def move_in_direction(direction: np.ndarray, distance: float):
    """Move the gripper in the given direction in a straight line by certain distance.
    """

def follow_way(path: List[Pose]):
    """Move the gripper to follow a path of poses."""

def rotate(angle: float)
    """Rotate the gripper clockwise at certain degree while maintaining the original position."""

def open_gripper():
    """Open the gripper to release the object, no args"""

def close_gripper():
    """Close the gripper to grasp object, no args. Move to best grasp pose before close gripper.
    """

---------------
Rules you have to follow:
#Directions: right: [0,1,0], left: [0,-1,0], upward or lift object: [0,0,1], forward or move
    away: [1,0,0]
#Please solve the following instruction step-by-step.
#You should ONLY implement the main() function and output in the Python-code style. Except
    the code block, output fewer lines.
---------------
Begin to excecute the task:
#Instruction:
```

Listing 2: An example of a full prompt in `RoboPro` with API renaming

```python
"""You're a vision language model controlling a gripper to complete manipulation tasks.
    Combine the images you see with the text instructions to generate detailed and workable
    code for the current scene.
You have access to the following tools:
"""
------------
import numpy as np
import torch
import math

#Perception Modules
def detect_bbox(description: str)->list[bbox]:
    """get the 2D boundingbox of all objects match description. When it comes to the specific
        parts or orientation of objects, the description should be detailed. Like 'handle of
        microwave', 'left side of shelf'.
    Return: list[bbox: np.ndarray]"""

def generate_obj_grasp_pos(grasp_bbox: bbox):
    """get best grasp pose to grasp specific object.
```

```
1026    Return: grasp_pose: Pose"""
1027
1028    def best_place_locator(holder_bbox: bbox):
1029    """Predict the place pose for an object relative to a holder
1030    Args: holder_bbox: bbox of target region of the holder.
        Return: place_pose: Pose"""
1031
1032    def find_axis_of_joint(joint_object_name: str):
1033    """Get the joint direction of an object
1034    Args: joint_object_name: the name of object have joint axis.
        Return: joint_axis: np.ndarray"""
1035
1036    def map_joint_path(joint_axis: np.ndarray, open: bool):
1037    """Generate a gripper path of poses around the joint. open is True when need open container
            around joint, False when close container.
1038    Return: path: list[Pose]
        """
1039
1040    def build_slide_path(target: Optional[str] = None, direction: Optional[np.ndarray] = None):
1041    """Generate path of poses to slide or push object to target or in specific direction.
1042    Args:
        target: The target location. If provided, 'direction' must be None.
1043    direction: The direction vector to slide the object along. If provided, 'target' must be None.
1044
        Return: path: list[Pose]
1045    """
1046
1047    def sweep_motion_path(object: Optional[str] = None, target: Optional[str] = None, direction:
            Optional[np.ndarray] = None):
1048    """This function is designed to generate movement paths for sweeping actions using tools such
1049        as sweepers, brooms. Grasp the tool before sweeping.
1050    Args:
        object: The object to be swept. If set to None, the function will perform a general sweeping.
1051    target: The target area or location to sweep towards. If provided, 'direction' must be None.
1052    direction: The direction vector for the sweeping motion. If provided, 'target' must be None.
        Return: path: list[Pose]
1053    """
1054
1055    def create_wipe_path(region: str):
1056    """This function is designed to generate movement paths for wiping actions using tools such
            as towel, sponge. Grasp the tool before wiping.
1057    Args:
1058    region (str): region to be wiped or cleaned.
        Return: path: list[Pose]
1059    """
1060
1061    def pour_path_mapper(grasped object: str, target: str):
1062    """Generate gripper path of poses to pour liquid in grasped object to target.
        Return: path: list[Pose]
1063    """
1064
1065    def best_press_pos(bbox):
        """Get best pose to press or push buttons."""
1066
1067    #Action Modules
1068    def relocate_to_pose(Pose):
        """Move the gripper to pose."""
1069
1070    def reach_in_direction(direction: np.ndarray, distance: float):
1071    """Move the gripper in the given direction in a straight line by certain distance.
        """
1072
1073    def follow_path(path: List[Pose]):
1074    """Move the gripper to follow a path of poses."""
1075
1075    def spin_gripper(angle: float)
1076    """Rotate the gripper clockwise at certain degree while maintaining the original position."""
1077
1078    def open_claw():
        """Open the gripper to release the object, no args"""
1079
        def clamp_gripper():
```

```
1080   """Close the gripper to grasp object, no args. Move to best grasp pose before close gripper.
1081       """
1082
1083   ---------------
1084   Rules you have to follow:
       #Directions: right: [0,1,0], left: [0,-1,0], upward or lift object: [0,0,1], forward or move
1085       away: [1,0,0]
1086   #Please solve the following instruction step-by-step.
       #You should ONLY implement the main() function and output in the Python-code style. Except
1087       the code block, output fewer lines.
1088   ---------------
1089   Begin to excecute the task:
       #Instruction:
1090
1091
1092
```

Listing 3: An example of a full prompt in `RoboPro` with API refactoring

```
1094   """You're a vision language model controlling a gripper to complete manipulation tasks.
1095       Combine the images you see with the text instructions to generate detailed and workable
1096       code for the current scene.
       You have access to the following tools:
1097   """
1098   ------------
1099   import numpy as np
       import torch
1100   import math
1101
1102   #Perception APIs
       def get_obj_bbox(description: str) -> list[np.ndarray]:
1103   """Get the 2D bounding box of all objects that match the description. The description should
1104       be detailed when it comes to specific parts or orientations of objects, such as 'handle
       of microwave' or 'left side of shelf'.
1105   Args:description (str): The description of the objects to find.
1106   Returns:list[np.ndarray]: A list of bounding boxes for the objects matching the description.
1107       """
1108
       def get_joint_axis(joint_object_bbox: np.ndarray):
1109   """Get the joint direction of an object
1110   Args: joint_object_name: the name of object have joint axis.
       Return: joint_axis: np.ndarray"""
1111
1112   #Control APIs
1113   def get_best_grasp_pos(grasp_bbox: np.ndarray):
       """Calculate the best grasp pose to grasp a specific object.
1114   Parameters: grasp_bbox (np.ndarray): The bounding box of the object to grasp.
1115   Return: Pose: The best grasp pose for the given object."""
1116
1117   def get_place_pos(holder_bbox: np.ndarray):
       """Predict the place pose for an object relative to a holder.
1118   Parameters: holder_bbox (np.ndarray): The bounding box of the target region of the holder.
1119   Return: Pose: The predicted place pose for the given object."""
1120
       def generate_joint_path(joint_axis: np.ndarray, open: bool) -> list[Pose]:
1121   """Generate a gripper path of poses around the joint.
1122   Parameters: joint_axis (np.ndarray): The axis of the joint. open (bool): True if the
       container needs to be opened around the joint, False if it needs to be closed.
1123   Returns: list[Pose]: The generated path of poses around the joint."""
1124
1125   def generate_slide_path(target_bbox: np.ndarray) -> List[Pose]:
       """Generate a path of poses to slide or push an object to a target or in a specific direction.
1126
1127   Parameters: target_bbox (np.ndarray): bbox of the target location.
1128   Returns: List[Pose]: The generated path of poses."""
1129
       def generate_sweep_path(target_bbox: np.ndarray) -> List[Pose]:
1130   """Generate movement paths for sweeping actions using tools such as sweepers or brooms. Grasp
1131       the tool before sweeping.
       Parameters: target_bbox (np.ndarray): The target area or location to sweep towards.
1132   Returns: List[Pose]: The generated path of poses for the sweeping action."""
1133
       def generate_wipe_path(region_bbox: np.ndarray) -> List[Pose]:
```

```
"""Generate movement paths for wiping actions using tools such as towels or sponges. Grasp
    the tool before wiping.
Parameters: region_bbox (np.ndarray): The region to be wiped or cleaned.
Return: List[Pose]: The generated path of poses for the wiping action."""

def generate_pour_path(grasped_object: str, target_bbox: np.ndarray) -> List[Pose]:
"""Generate a gripper path of poses to pour liquid from a grasped object to a target.
Parameters: grasped_object (str): The object being grasped that contains the liquid.
    target_bbox (np.ndarray): The bounding box of the target area where the liquid will be
    poured.
Returns: List[Pose]: The generated path of poses for the pouring action."""

def generate_press_pose(bbox: np.ndarray) -> Pose:
"""Get the best pose to press or push buttons.
Parameters: bbox (BBox): The bounding box of the button or area to be pressed.
Return: Pose: The best pose for pressing or pushing the button."""

def move_to_pose(pose: Pose):
"""Move the gripper to the specified pose.
Parameters: pose (Pose): The target pose to move the gripper to."""

def move_in_direction(direction: np.ndarray, distance: float):
"""Move the gripper in the given direction in a straight line by a certain distance.
Parameters: direction (np.ndarray): The direction vector to move the gripper along. distance
    (float): The distance to move the gripper."""

def follow_way(path: List[Pose]) -> None:
"""Move the gripper to follow a path of poses.
Parameters: path (List[Pose]): The list of poses that defines the path to follow."""

def rotate(angle: float) -> None:
""" Rotate the gripper clockwise by a certain angle while maintaining the original position.
Parameters: angle (float): The angle in degrees to rotate the gripper."""

def open_gripper() -> None:
"""Open the gripper to release the object.
Parameters: None
Returns: None"""

def close_gripper() -> None:
"""Close the gripper to grasp an object. Move to the best grasp pose before closing the
    gripper.
Parameters: None
Returns: None"""

---------------
Rules you have to follow:
#Directions: right: [0,1,0], left: [0,-1,0], upward or lift object: [0,0,1], forward or move
    away: [1,0,0]
#Please solve the following instruction step-by-step.
#You should ONLY implement the main() function and output in the Python-code style. Except
    the code block, output fewer lines.
---------------
Begin to execcute the task:
#Instruction:
```