# OpenReview forum: "Robotic Programmer: Video Instructed Policy Code Generation for Robotic Manipulation"
_ICLR.cc/2025/Conference — Submitted to ICLR 2025_

### Official Review · Reviewer_R1W2 · 2024-11-02

**Soundness:** 2
**Presentation:** 1
**Contribution:** 2
**Rating:** 3
**Confidence:** 4

**Summary:**

This work proposes RoboPro, a high-level planner that generates sequences of low-level API calls based on language instructions and observations. It leverages a two-stage approach, using a Vision-Language Model (VLM) and a code Language Model (code LLM) to process videos and generate runtime code data, which is then used for fine-tuning RoboPro. Experiments cover nine tasks from RLBench, eight tasks from LIBERO, and several real robot tasks, demonstrating RoboPro's zero-shot planning generalization when low-level skills and APIs change.

**Strengths:**

This work processes videos to generate robot planning data using a two-stage approach.
The experiments demonstrate the model's planning generalization ability across two benchmarks.

**Weaknesses:**

1. The contributions of this work are not clearly summarized.
2. The writing could be improved, as there are many redundant sentences that convey the same ideas. The structure and presentation are somewhat difficult to follow.
3. The implementation of low-level skills is not well-explained for both simulation and real work experiments. The assumption of predefined API calls and low-level skills is too strong for developing effective manipulation policies.
4. The experiments are primarily limited to low-precision, open-loop pick-and-place tasks.

**Questions:**

Since RoboPro is pre-trained on related robot runtime code data, were the baseline planners also fine-tuned on this data?

Peract is a multi-task policy learned from demonstrations, directly outputting the keyframe action without a high-level planner. How is it an apples-to-apples comparison with RoboPro which uses a high-level planner equipped with low-level skills?

RoboPro is referred to as a generalist. However, popular generalist baselines like RT-2[1] and Open-VLA[2] are missing from the comparisons.

Can RoboPro complete tasks when a target is moving during planning? Is there some way to address it?

Could you briefly explain how RoboPro might be extended to solve deformable object manipulation tasks, such as folding cloth?

What is the main difference between RoboPro and RoboCodeX, aside from the pre-training on robot runtime data?

[1] RT-2: Vision-Language-Action Models Transfer Web Knowledge to Robotic Control
[2] Open-VLA: An Open-Source Vision-Language-Action Model

---

> ### Author Response · Authors · 2024-11-21
>
> Thanks for feedback of the reviewer. We'll continuously polish our paper and improve our writing. We also clarified our main contributions again in the Global Rebuttal (https://openreview.net/forum?id=baQ0ICrnCR&noteId=uooLlWYLvE).
>
> > **Weakness 3: Implementation of low-level skills**
>
> We appreciate for the reviewer’s feedback. Both simulation and real-world experiments adopt the same low-level skills implementation. Due to the space limitation of the main paper, we have provided information regarding the division of the API library, as well as input and output formats in **Sec. 3.1**. Additionally, we’ve illustrated some details about the modules used for implementation of the low-level skills in **Sec. 3.2 \(L256-L260\)**. As addressed in **Sec. 4.1 \(L311\)**, the detailed implementation of our API library is demonstrated in **Appendix B**. Please refer to the corresponding sections of our paper for more specific details.
>
> > **Weakness 3: The assumption of predefined API calls and low-level skills**
>
> The fundamental assumption of predefined API calls and low-level skills is a widely accepted premise in policy code generation approaches[1][2]. By integrating skills across various categories and hierarchies, these methods enable the execution of exponentially compositional tasks.
>
> We need to clarify that our primary contribution is NOT to develop comprehensive skills for all types of tasks, but to introduce an efficient pipeline for synthesizing runtime code from extensive in-the-wild videos and a robotic foundation model to perceive visual information and follow free-form instructions in a zero-shot manner.
>
> Generalization ability out of predefined skill libraries has been largely under-explored in prior policy code generation works. As mentioned in our Global Rebuttal (https://openreview.net/forum?id=baQ0ICrnCR&noteId=uooLlWYLvE), we make an early effort to investigate zero-shot generalization capabilities of policy code generation across different APIs and skills, extending beyond the constraints of the predefined API set.
>
> As discussed in **Sec. 4.2 \(L398-L410\)**, we've validated RoboPro's robustness to variations in formation of the API library, such as API Renaming and API Refactoring. To further evaluate RoboPro's adaptability to newly defined or task-specific APIs, we selected three compositional tasks from RLBench that involve multi-step execution: **Water Plants**, **Hit Ball** and **Scoop Cube**. For each task, we designed a new set of task-specific APIs encompassing skills not included in RoboPro's training phase. As shown in **Tab.1 of our Global Rebuttal**, the performance of RoboPro consistently outperforms GPT-4o and CaP in a zero-shot manner. This robustness is attributed to RoboPro's ability as a generalist on code generation to comprehend newly defined functions and sequential action knowledge learned from Video2Code. This combination enables RoboPro to seamlessly adapt to evolving API structures and new task demands, thus offering a flexible and efficient solution for robotic manipulation in various environments.
>
> Future works orthogonal to our focus in this study can discuss how to construct the skill library in a dynamically extensible manner by making the API library learnable.
>
> > **Weakness 4: The experiments are low-precision, open-loop pick-and-place tasks.**
>
> This is not true. We would like to argue that most of our evaluation tasks are NOT low-precision, open-loop pick-and-place tasks. The evaluation of RoboPro contains 9 tasks in RLBench, 8 tasks in LIBERO, and 8 tasks in real environments. These carefully designed tasks include evaluation on novel skills (wipe, sweep, push, slide, rotate, directional moving), long-termed instructions (multi-round stacking, pushing, pick-placing, manipulation of articulated objects), tool-using (sponge, sweeper, buttons, watering can), and visual grounding ability (visual preference, identification, vague instructions). Prior works such as Instruct2Act[3] is limited to 2D scenarios and suction-based pick-and-place tasks, while RoboCodeX[2] employs custom-built environments rather than standardized simulation platforms. On **widely-used** simulators and real-world environments, the evaluation of RoboPro has tried the best to cover most primary tasks evaluated in previous works while enriching the diversity and complexity of evaluation scenarios and skills.
>
> Reference
>
> [1] Liang, Jacky, et al. Code as policies: Language model programs for embodied control, ICRA, 2023.
>
> [2] Mu, Yao, et al. RoboCodeX: Multimodal Code Generation for Robotic Behavior Synthesis, NeurIPS, 2024.
>
> [3] Huang, Siyuan, et al. Instruct2act: Mapping multi-modality instructions to robotic actions with large language model, arXiv, 2023.

---

> ### Author Response · Authors · 2024-11-21
>
> > **Question 1: Were the baseline planners also fine-tuned on this data?**
>
> The baseline code generation methods, such as GPT-4o and CaP, utilize closed-source large language models. The training sets of these models are unclear. We didn’t fine-tune these models on the data generated by Video2Code, which is also publicly unavailable or extremely expensive. Using closed-source models without fine-tuning as baselines is common practice in this field[1]. It is also worth noting that these baseline models have more than 10 times parameters compared to RoboPro. We’re exploring efficient method for the training of open source VLMs on policy code generation.
>
> > **Question 2: How is it an apples-to-apples comparison with RoboPro which uses a high-level planner equipped with low-level skills?**
>
> It is rather difficult to ensure absolute fairness for the comparison of code generation approaches and behavior cloning (BC) methods. Besides the output format, code generation methods are evaluated in a zero-shot way, while BC methods need additional fine-tuning to show reasonable performance. Previous works in this field primarily make comparisons with other code generation methods like CaP or closed-source models like GPT-4V. Such standard comparisons have been included in our evaluation across different tasks and environments. We have also provided extra comparisons with behavior cloning (BC) methods, such as PerAct, primarily as a reference for upper bound performance on existing benchmarks. We observe that the zero-shot performance of RoboPro surpasses other code generation baselines and is favorably comparable to BC methods trained on evaluation tasks.
>
> > **Question 3: Comparisons with more VLAs**
>
> RT-2 and OpenVLA can be categorized into vision-language action models (VLAs). For VLAs, we compared our method with concurrent work LLARVA in **Sec. 4.1 \(L330-L331\)**, which shares a similar technical approach with OpenVLA. Despite these models achieve better performance and show capacity to transfer on novel objects and different tasks, fine-tuning is still required when deploying on new robots and environments.
>
> Since RT-2 is a closed-source model, during the rebuttal, we conducted a further comparison with OpenVLA on LIBERO. We report success rate of OpenVLA on 8 LIBERO tasks with two different settings: **zero-shot** performance as a generalist, and **fine-tuning** on all evaluation tasks as a reference for training-based methods. As shown in **Tab.1.**, when evaluated in a zero-shot setting, OpenVLA demonstrated no measurable performance, while the zero-shot performance of RoboPro is marginally comparable with OpenVLA fine-tuned on all evaluation tasks. The overall experimental results are consistent with those of other behavior cloning (BC) methods, such as PerAct. We will supplement the results of OpenVLA in our revision.
>
> **Table 1. Success Rate on 8 LIBERO Tasks Compared with OpenVLA.**
>
> |  | **Turn on Stove** | **Close Cabinet** | **Put in Sauce** | **Put in Butter** | **Put in Cheese** | **Place Book** | **Boil Water** | **Identify Plate** | **Average** |
> |--------------------------|-------------------|-------------------|------------------|-------------------|-------------------|----------------|----------------|--------------------|-------------|
> | **OpenVLA (fine-tuned)** | 97.0             | 97.0             | 37.0            | 60.0             | 53.0             | 93.0          | 43.0          | 40.0              | 65.0        |
> | **OpenVLA (zero-shot)**  | 0.0              | 0.0              | 0.0             | 0.0              | 0.0              | 0.0           | 0.0           | 0.0               | 0.0         |
> | **RoboPro (zero-shot)**  | 97.0             | 60.0             | 67.0            | 53.0             | 63.0             | 43.0          | 23.0          | 13.0              | 52.4        |
>
> **Reference**
>
> [1] Mu, Yao, et al. RoboCodeX: Multimodal Code Generation for Robotic Behavior Synthesis, NeurIPS, 2024.

---

> ### Author Response · Authors · 2024-11-21
>
> > **Question 4 & 5: Moving and deformable targets**
>
> We thanks the reviewer’s questions for further discussion and we would like to clarify again that the key contribution of this work is using action-free data videos to synthesize robotic runtime code, which overcomes the data bottleneck for the training of robotics foundation models to generate visual-grounded policy code. How to manipulate on moving and deformable targets is an interesting and open question for robotic manipulation, however, it is beyond the scope of this study. These questions are more relevant with high-frequency responsive controllers and dexterous manipulation with tactile sensing and not the main focus of our paper. As suggested by Reviewer 9oqZ, RoboPro can serve as a planner for moving targets by incorporating intermediate frames. Leveraging intermediate frames is now limited to interleaved training of existing VLMs, the frequency of controllers and robotics hardware, which deserves further exploration.
>
> > **Question 6: Differences with RoboCodeX**
>
> As summarized in our Global Rebuttal (https://openreview.net/forum?id=baQ0ICrnCR&noteId=uooLlWYLvE), the key contribution of this paper is Video2Code, a scalable and efficient multimodal code generation pipeline from demonstration videos. To the best of our knowledge, we make an early attempt to train **end-to-end** policy code generation model to perceive visual information and follow free-form instructions, which is RoboPro. Previous work RoboCodeX employs a two-stage approach to first use VLM to generate high-level plans and preferences, and then translate these textual outputs to generated policy code.
>
> The unified pipeline for perception, instruction following and coding in an end-to-end fashion can effectively eliminate the potential loss of critical information during intermediate steps and enhance computational efficiency during inference. However, training such VLMs will inevitably require a vast amount of diverse and well-aligned robot-centric multimodal runtime code data, which poses a significant challenge. To mitigate this challenge, Video2Code plays the most important role to make the training of RoboPro feasible and effective. This scalable and automatic data curation pipeline directly aligns code with visual information and procedural knowledge from video demonstrations, significantly enhancing scalability and quality of generated policy code.
>
> Extensive and comprehensive experiments have verified the effectiveness of RoboPro on both simulation environments and real-world settings. RoboCodeX was not directly compared in our experiments as this work has not released its model checkpoints and its original evaluations were not conducted on publicly available simulation platforms.

---

> > ### Comment · Reviewer_R1W2 · 2024-11-26
> >
> > Thank you for your response. While some of my questions have been addressed, I still have concerns regarding the structure, writing, and experiments. To clarify, Robopro should be considered a code-generation planning method rather than a generalist policy. While the new planning data effectively enhances planning capabilities, it seems unfair to compare this method with ChatGPT, which lacks in-context learning, or with other code-generation planning methods that have not been fine-tuned on this specific dataset, as well as with an LfD method. Consequently, it is difficult to conclude that the proposed method functions as a generalist policy when compared to these baselines. I recommend that the authors reorganize their contributions to emphasize the significance of the dataset and its potential to improve a wide range of planning methods.

---

> > > ### Author Response · Authors · 2024-11-28
> > > **Further Explanation**
> > >
> > > Thanks for your reply. We are glad that our response has resolved some of your questions. We would like to further address your remaining concerns and possible confusions.
> > >
> > > The most relevant baselines are other policy code generation methods, i.e., GPT-4o and CaP. Both GPT-4o and
> > > Cap leverage closed-source foundation models, such as GPTs, which feature over 10 times the parameters and are pretrained on substantially larger datasets compared to our RoboPro.
> > > However, these advanced proprietary LLMs face significant limitations, including high computational costs, and reliance on specialized data not available to the public. Moreover, fine-tuning to include domain knowledge and enhance capabilities is not feasible for these proprietary models. To tackle these issues, RoboPro has proposed to explore another line of approaches to effectively fine-tune open-source VLMs, which we believe is a more promising way to enhance performance on specific areas (i.e., policy code generation) and eliminate the dependence on advanced proprietary LLMs like ChatGPT. The motivation of Video2Code, our data curation pipeline, also lies in effectively extracting procedural knowledge from videos to facilitate the training of open-source models.
> > >
> > > Using closed-source models without fine-tuning as baselines is common practice both for code generation methods[1] and general tool-using[3] in this field. To **make every effort to ensure the fairness of the comparison**, we have exactly **used in-context learning** during the evaluation of GPT-4o and Cap. (from your response, you probably have misunderstanding on this point, and we'd like to make it clear here). As shown in Sec 4.1 (L309-L311),
> > > for a fair comparison with other approaches using closed-source models, we adopt an evaluation method similar to that in RoboCodeX[1], where detailed explanations and definitions of functions and skills are provided in
> > > the prompt. This has proven to be a more effective approach for in-context learning of policy code generation[2].
> > > Corresponding prompts for evaluation can be found in **Appendix B**.
> > >
> > > For LfD methods (PerAct), it is rather difficult (if not impossible) to ensure completely fairness for comparisons between code generation approaches and
> > > behavior cloning (BC) methods. As stated in our paper, such training-based methods are primarily used as a reference upper bound for existing benchmarks. We observe that
> > > the zero-shot performance of RoboPro is favorably comparable to BC methods trained on evaluation tasks, and shows better generalization ability across tasks
> > > and environments.
> > >
> > > RoboPro is trained on real-world data, and evaluated on simulators to compare with other methods. To further guarantee fairness in
> > > comparisons, we also carefully verified that scenes and instructions are completely unseen during the test phase. Under these settings,
> > > we have successfully demonstrated RoboPro's zero-shot generalization ability on tasks, instructions, and environments.
> > >
> > > **Reference**
> > >
> > > [1] Mu, Yao, et al. RoboCodeX: Multimodal Code Generation for Robotic Behavior Synthesis, NIPS, 2024.
> > >
> > > [2] Huang, Siyuan, et al. Instruct2act: Mapping multi-modality instructions to robotic actions with large language model, arXiv, 2023.
> > >
> > > [3] Yang, Rui, et al. Gpt4tools: Teaching large language model to use tools via self-instruction, NIPS, 2024.

---

### Official Review · Reviewer_p96h · 2024-11-03

**Soundness:** 2
**Presentation:** 3
**Contribution:** 2
**Rating:** 3
**Confidence:** 4

**Summary:**

The paper presents RoboPro which enables zero-shot performance on robot manipulation tasks by converting observations and language instructions into executable policy code. RoboPro's code generation is enhanced by Video2Code, a data curation pipeline that outputs executable code from input videos.

**Strengths:**

1.  Experiments are comprehensive, with both simulation and real-world tasks, as well as general VQA tasks.

**Weaknesses:**

1. Performance on more complex long-horizon tasks is not thoroughly explored.

2. RoboPro depends on consistent API libraries. It's unclear how this method scales with open-ended real-world tasks of arbitrary complexity.

**Questions:**

1. Can the authors provide additional contextualization in lines 122-136 with prior works such as [1]?

2. I'm also not sure how scalable this method actually is. Certainly the generalizability of LLMs is well-leveraged, but the interface between various components seems like a bottleneck. Can the authors explain how the code API returned by Video2Code can be adapted if new functions are required to describe tasks in new video demonstrations? It seems like changing the code API requires re-running the Video2Code pipeline on the entire video dataset?

[1] [Programmatically Grounded, Compositionally Generalizable Robotic Manipulation](https://arxiv.org/abs/2304.13826)

---

> ### Author Response · Authors · 2024-11-21
>
> We would like to thank you for your insightful comments and suggestions. We have carefully considered each point raised and have provided detailed responses below.
>
> > **Weakness 1: Performance on more complex long-horizon tasks is not thoroughly explored.**
>
> Thanks for your comments. We'd like to clarify that RoboPro has tried the best to cover most primary tasks evaluated in previous works while enriching the diversity and complexity of evaluation scenarios and skills on evaluation tasks.
>
> Prior works such as Instruct2Act[2] is limited to 2D scenarios and suction-based pick-and-place tasks, while RoboCodeX[3] employed custom-built environments rather than standardized simulation platforms. Their evaluation tasks were primarily restricted to pick-and-place or articulated object manipulation, with multi-round tasks largely confined to multi-turn pick-and-place operations, lacking scenarios involving tool usage and other types of actions.
>
> Due to the limitation of current existing benchmarks, we have made every effort to maximize the diversity of the evaluation tasks. The evaluation of RoboPro contains 9 tasks in RLBench, 8 tasks in LIBERO, and 8 tasks in real-world environments. These tasks include multi-round stacking, tool-using, pushing, manipulation of articulated objects, and etc. With more complicated long-horizon tasks being introduced in upcoming benchmarks, we are happy to evaluate them in the future work.
>
> > **Weakness 2 & Question 2: Scalability of our method with open-ended real-world tasks of arbitrary complexity and new functions required to describe tasks in new demonstrations.**
>
> Thanks for your comments. We would like to clarify that RoboPro does not rely on consistent API libraries. As mentioned in Global Rebuttal (https://openreview.net/forum?id=baQ0ICrnCR&noteId=uooLlWYLvE), this issue is under-explored in prior policy code generation works, as they all work under the assumption that a predefined API library is provided. We have made efforts to validate the scalability of code generation methods, by examining RoboPro's robustness to variations in API structures, such as API Renaming and API Refactoring. Additionally, we further evaluate RoboPro's adaptability to newly defined or task-specific APIs. As shown in **Tab. 1. in the main paper** and **Tab. 1. from the Global Rebuttal**, when encountering new or modified APIs during evaluation, RoboPro still demonstrates strong zero-shot adaptation capabilities. This allows it to effectively handle altered or newly defined skills without additional fine-tuning when extending to open-ended real-world tasks of arbitrary complexity.
>
> > **Question 1: Can the authors provide additional contextualization in lines 122-136 with prior works such as [1]**
>
> Thank you for your suggestion! The reference you suggested are beneficial for us. We have read the reference and learnt a lot. PROGRAMPORT[1] utilizes a Combinartory Categorial Grammar (CCG) to parse the sentence into a “manipulation program”, based on a compact but general domain-specific language (DSL), which enables directly leverage of a pretrained vision language (VL) model and disentangles the learning of visual grounding and action policies. Robot-centric policy code generation methods replace DSL with executable code which is more suitable for VLMs to process. The reference you suggest enriches our discussion of related works on using code generation methods to solve robotic manipulation tasks, by presenting an alternative approach to bridge vision-language models with low-level actions beyond the use of executable code, which is helpful for improving the quality of the article. We will cite it in the revised manuscript. Thanks again.
>
> **Reference**
>
> [1] Wang, Renhao et al. Programmatically Grounded, Compositionally Generalizable Robotic Manipulation, ICLR, 2023.
>
> [2] Huang, Siyuan, et al. Instruct2act: Mapping multi-modality instructions to robotic actions with large language model, arXiv, 2023.
>
> [3] Mu, Yao, et al. RoboCodeX: Multimodal Code Generation for Robotic Behavior Synthesis, NeurIPS, 2024.

---

### Official Review · Reviewer_xsvW · 2024-11-04

**Soundness:** 3
**Presentation:** 3
**Contribution:** 2
**Rating:** 5
**Confidence:** 4

**Summary:**

This paper presents RoboPro (Robotic Programmer), a robotic foundation model that generates executable policy code from visual information and language instructions to perform robot manipulation tasks without additional fine-tuning. The authors propose Video2Code, an automatic data curation pipeline that synthesizes code execution data from large-scale instructional videos. Extensive experiments are conducted in both simulators and real-world environments to validate the model's effectiveness. In short, the reviewer thinks this paper has good presentation, thorough evaluations, but lacks novelty and insights.

**Strengths:**

1. The Video2Code pipeline bridges the gap between visual understanding and code generation by combining the visual reasoning ability of VLMs and the coding proficiency of code-domain LLMs. This low-cost and automatic method reduces reliance on manually constructed datasets and expensive simulation environments.

2. RoboPro shows adaptability to different API configurations and compatibility across environments (simulators like RLBench and LIBERO, as well as real-world settings), underscoring its robustness and usability in diverse practical scenarios.

3. Extensive experiments in simulations and real-world scenarios verify the model's code generation abilities, with analysis on how different code LLMs affect performance.

**Weaknesses:**

1.The auto code data collection is intuitive and simple, which does not count as "novel" for the VLM agent training. The auto code collection pipeline is natural by itself and has been adopted in many applications like multi-modal OS agents and game agents. However, since you have real-time feedback from the real worlds, if would be of more interesting how you could accelerate this data collection and enhance the code quality from the multi-modal reflection on the world feedback.

2.The use of a foundation model for code generation lacks methodological innovation, as has been cited by the authors, there are already published papers that can fall into this category. The authors should try to highlight the difference of this work among other reference methods.

3.The model's zero-shot capability is restricted by the predefined API library it can call.

**Questions:**

1. Since the authors have conducted comprehensive experiments, maybe they can share more insight about the limitations of code generation approaches compared to VLAs? By comparing the difference between these two approaches, it would be more interesting for the authors to share some insights about what can be done or what cannot be done by their approach and where direction we could go for the future work.

2. Include a more detailed analysis of failure cases, distinguishing between issues related to LLM reasoning and API limitations. This could provide more insight for this paper. It's unclear whether failures are due to issues with LLM Chain-of-Thought reasoning or problems with the API itself, etc.

---

> ### Author Response · Authors · 2024-11-21
>
> We would like to thank you for your insightful comments and suggestions. We have carefully considered each point raised and have provided detailed responses below.
>
> > **Weakness 1: The auto code data collection is intuitive and simple, which does not count as "novel" for the VLM agent training. The auto code collection pipeline is natural by itself and has been adopted in many applications like multi-modal OS agents and game agents. However, since you have real-time feedback from the real worlds, if would be of more interesting how you could accelerate this data collection and enhance the code quality from the multi-modal reflection on the world feedback.**
>
> We appreciate the reviewer's feedback and would like to clarify the novelty and distinctiveness of our auto code data collection approach, particularly for robotic applications. As mentioned by the reviewer, OS agents are implemented in virtual world and limited action space. Thus, using random sampling (MobileVLM[1]) or making up examples by LLMs (WebSight[2]) are direct ways for applications in these areas. However, for robotic models, sampling based pipelines are with high cost for real-world embodiments. Synthesizing and verifying in simulation environments lacks variations and requires extra efforts for manually-built environments.
>
> To handle these problems, we devise Video2Code, an automatic and scalable data curation pipeline using action-free videos to synthesize robotic runtime code. Our approach is inspired by the vast repository of in-the-wild operational videos, which inherently capture procedural knowledge required to accomplish a wide range of tasks. These videos serve as high-quality demonstrations of task execution following free-form language instructions. However, these demonstration videos lack corresponding policy code annotations. Our contribution lies in the design of a novel two-stage automated data pipeline that transforms these rich video demonstrations into executable robotic code. To the best of our knowledge, we are the first to leverage unstructured video content as a source for generating runtime code data in robotics, which has significantly reduced the cost and enhanced the efficiency of data curation for policy code generation. Furthermore, for robotics demonstrations, we could utilize pose information that is either pre-recorded or can be extracted from videos. This pose information is instrumental in identifying key frames, thereby accelerating the automated data curation process.
>
> > **Weakness 2: The use of a foundation model for code generation lacks methodological innovation, as has been cited by the authors, there are already published papers that can fall into this category. The authors should try to highlight the difference of this work among other reference methods.**
>
> The key contribution of this paper is Video2Code, a scalable and efficient multimodal code generation pipeline from demonstration videos. To the best of our knowledge, we make an early attempt to train end-to-end policy code generation model to perceive visual information and follow free-form instructions, which is RoboPro. Previous work RoboCodeX[3] employs a two-stage approach to first use VLM to generate high-level plans and preferences, and then translate these textual outputs to policy code. As stated in **Sec. 2 (L122-L130)**, other approaches primarily fall into prompting closed-source linguistic models[4][5].
>
> The unified pipeline for perception, instruction following and coding in an end-to-end fashion can effectively eliminate the potential loss of critical information during intermediate steps and enhance computational efficiency during inference. However, training such VLMs will inevitably require a vast amount of diverse and well-aligned robot-centric multimodal runtime code data, which poses a significant challenge. To mitigate this challenge, Video2Code plays the most important role to make the training of RoboPro feasible and effective. This scalable and automatic data curation pipeline directly aligns code with visual information and procedural knowledge from video demonstrations, significantly enhancing scalability and quality of generated policy code.
>
> **Reference**
>
> [1] Wu, Qinzhuo, et al. MobileVLM: A Vision-Language Model for Better Intra-and Inter-UI Understanding, 2024.
>
> [2] Laurençon, Hugo, et al. Unlocking the conversion of Web Screenshots into HTML Code with the WebSight Dataset, 2024.
>
> [3] Mu, Yao, et al. RoboCodeX: Multimodal Code Generation for Robotic Behavior Synthesis. NIPS. 2024.
>
> [4] Liang, Jacky, et al. Code as policies: Language model programs for embodied control, ICRA, 2023.
>
> [5] Huang, Siyuan, et al. Instruct2act: Mapping multi-modality instructions to robotic actions with large language model, arXiv, 2023.

---

> ### Author Response · Authors · 2024-11-21
>
> > **Weakness 3: The model's zero-shot capability is restricted by the predefined API library it can call.**
>
> Generalization ability out of predefined skill libraries has been largely under-explored in prior policy code generation approaches. As mentioned in our Global Rebuttal (https://openreview.net/forum?id=baQ0ICrnCR&noteId=uooLlWYLvE), we make an early effort to investigate zero-shot generalization capabilities of policy code generation across different APIs and skills, extending beyond the constraints of the predefined API set.
>
> As discussed in **Sec. 4.2 \(L398-L410\)**, we've validated RoboPro's robustness to variations in formation of the API library, such as API Renaming and API Refactoring. To further evaluate RoboPro's adaptability to newly defined or task-specific APIs, we selected three compositional tasks from RLBench that involve multi-step execution: **Water Plants**, **Hit Ball** and **Scoop Cube**. For each task, we designed a new set of task-specific APIs encompassing skills not included in RoboPro's training phase. As shown in **Tab.1. of our Global Rebuttal**, the performance of RoboPro consistently outperforms GPT-4o and CaP in a zero-shot manner. This robustness is attributed to RoboPro's ability as a generalist code model to comprehend newly defined functions and sequential action knowledge learned from Video2Code. This combination enables RoboPro to seamlessly adapt to evolving API structures and new task demands, thus offering a flexible and efficient solution for robotic manipulation in various environments.
>
> Future works orthogonal to our focus in this study can discuss how to construct the skill library in a dynamically extensible manner by making the API library learnable.
>
> > **Question 1: Since the authors have conducted comprehensive experiments, maybe they can share more insight about the limitations of code generation approaches compared to VLAs? By comparing the difference between these two approaches, it would be more interesting for the authors to share some insights about what can be done or what cannot be done by their approach and where direction we could go for the future work.**
>
> We thanks the reviewer's constructive suggestion. To make a further analysis, we provide an extra comparison with OpenVLA on LIBERO in Rebuttal (https://openreview.net/forum?id=baQ0ICrnCR&noteId=0ny6CIE7QG). VLAs achieve better performance and show the capacity to transfer on novel objects and different tasks, while fine-tuning is still required when deploying on new environments. Code generation approaches emphasize generalization and compositional reasoning, demonstrating zero-shot generalization ability across
> environments and tasks. The bottlenecks of current policy code generation systems are twofold: the reasoning capabilities of code generation models and the precision of low-level skills.
>
> In this paper, our main focus is the reasoning and generalization ability of the policy code generation model. With scalable and automatic data curation pipeline Video2Code, we enable RoboPro to perceive visual information and follow free-form instructions in a zero-shot manner. As shown in extensive experiments and error breakdown in Figure 4, policy code generation systems equipped with RoboPro have effectively reduced failure cases in grounding and reasoning.
>
> Another major performance bottleneck, orthogonal to our focus, lies in the implementation of low-level APIs. Better detection models and grasping models can both enhance the performance of code generation approaches. At the same time, the atomic skills required by embodiments to complete real-world tasks are not infinite. By combining skills from different categories and hierarchies, exponentially complex tasks can be accomplished, which is also a core idea of code generation methods. Discussing the completeness of skill libraries for different types of embodiments would be a feasible yet challenging direction for further exploration.

---

> ### Author Response · Authors · 2024-11-21
>
> > **Qestion 2: Include a more detailed analysis of failure cases, distinguishing between issues related to LLM reasoning and API limitations. This could provide more insight for this paper. It's unclear whether failures are due to issues with LLM Chain-of-Thought reasoning or problems with the API itself, etc.**
>
> We appreciate the reviewer's suggestion. An error breakdown on RLBench is depicted in Figure 4. As stated in **Sec. 4.1 (L356-L368)**, for policy code generation methods, the successful execution of manipulation tasks relies on both the accuracy of the policy code and the capabilities of the API library. The main types of errors caused by the reasoning ability of VLMs or LLMs are summarized as three different types: logical errors, functional errors, and grounding errors. These errors are associated with challenges in the appropriate selection and utility of APIs, as well as issues related to visual grounding. While the problems with the API itself refer to execution errors colored grey in Figure 4 , which maintain a consistent proportional relationship with successful cases. The results show that all these methods perform well on following functional definition of the API library, causing a low occupancy of functional error. Compared with linguistic only method CaP, GPT-4o and RoboPro show a noticeable improvement in target object grounding. The main failure cases of CaP and GPT-4o fall in logical error, including API selection and proper order of API calls. In contrast, RoboPro effectively reduces this margin, mainly owing to the procedural knowledge about long-term execution learned in Video2Code.

---

### Official Review · Reviewer_9oqZ · 2024-11-05

**Soundness:** 3
**Presentation:** 3
**Contribution:** 3
**Rating:** 6
**Confidence:** 4

**Summary:**

This paper proposes a method, RoboPro, that utilizes action-free video for zero-shot policy code generation in robotic manipulation tasks. The method consists of two components: a video-to-code model that generates robotic runtime code, and a code-generation policy trained on the synthesized code. In experiments, RoboPro was shown to be effective in performing tasks in unseen environments in a zero-shot manner.

**Strengths:**

1. The idea and motivation of using action-free data videos to synthesize robotic runtime code, which was then used to train a code policy is novel and interesting.

2. The experimental results demonstrate that RoboPro can perform the task in a zero-shot manner, which is compelling.

**Weaknesses:**

1. [Major] While the paper compares its approach to BC-based and code-generation methods, I’m curious how it performs compared to recent methods that use VLM prompting without model training, like MOKA [1], which leverages VLM reasoning and motion primitives, and PIVOT [2], which uses iterative visual prompting. These methods generally make fewer computational assumptions than RoboPro since they don’t require any module training.


2. [Major] In the real-world experiments, there is no comparison to other baselines. Would it be possible to include a comparison against at least GPT-4o to see if RoboPro performs better?


3. [Major] While the idea of using action-free video to synthesize runtime code data is interesting, the impact of the chosen video datasets remains unclear. In this paper, the authors use DROID as their dataset—was there any specific reason for this choice? How do the type and size of the dataset affect the final performance? If I’m understanding correctly, in principle, the video data doesn’t even need to be robotic. How would the approach perform if human videos, such as Ego4D or Something-Something, were used instead? It would be interesting to include the ablation over these points.


4. [Minor] The downstream performance appears to be sensitive to the choice of VLMs used for draft/code generation, as shown in Table 6. Specifically, there is a large difference in results between using Gemini and DeepSeek for code generation. This sensitivity to the VLM choice may be a weakness of this approach.

---
Reference

[1] Fang et al., MOKA: Open-World Robotic Manipulation through Mark-Based Visual Prompting, 2024.

[2] Nasiriany et al., PIVOT: Iterative Visual Prompting Elicits Actionable Knowledge for VLMs, 2024.

**Questions:**

1. How accurate is the generated code from the data curation pipeline? Are there any errors or failure cases and how are they handled?


2. Could the authors provide more details on the training procedure? – e.g. hyperparameters, training steps, and the computational resources and time required for training, etc


3. During the deployment phase, is it correct that only the initial image of the environment is used to generate the entire execution code, without using the newest images at each time step?

---

> ### Author Response · Authors · 2024-11-21
>
> We would like to thank you for your insightful comments and suggestions. We have carefully considered each point raised and have provided detailed responses below.
>
> > **Weakness 1: While the paper compares its approach to BC-based and code-generation methods, I’m curious how it performs compared to recent methods that use VLM prompting without model training, like MOKA [1], which leverages VLM reasoning and motion primitives, and PIVOT [2], which uses iterative visual prompting. These methods generally make fewer computational assumptions than RoboPro since they don’t require any module training.**
>
> Thanks for your suggestions. We'd like to clarify that our primary comparisons focus on the code generation methods, demonstrating that open-source models with well-aligned and high-quality data generated by Video2Code could achieve superior policy code generation capabilities compared to large closed-source models, like GPT-4o. The comparison with BC-based methods with additional fine-tuning serves as an upper-bound reference for performance on existing benchmarks, illustrating that our model achieves comparable results with training-based methods while highlighting the zero-shot generalization capability. Therefore, we did not include comparisons with visual prompting methods in our experiments.
>
> Although VLM prompting methods make fewer computational assumption as they don’t require any module training but directly rely on proprietary models, like GPT-4o, they typically require multiple rounds of queries for each subtask to achieve relatively precise planning for manipulation tasks, which is highly susceptible to the influence of hallucinations, especially for long horizon tasks, as failure in any subtask can result in the failure of the entire task. Besides, VLM prompting methods lack the perception regarding the robot’s embodiment, including what the robot can or cannot do. For example, perceiving the regions of task-relevant object is implemented using manually crafted, fixed inference code in **MOKA [1]**, instead of being called by robot itself, which would bring some challenges such as distinguishing between different individuals of the same class.
>
> As suggested, we conduct further experiments on **MOKA [1]** following the same real-world setting illustrated in the main paper and provide experiment results in **Tab. 1.**. It can be seen that RoboPro outperforms **MOKA [1]** in most of the tasks, especially for long-horizon tasks. For example, in **Prepare Meal**, the manipulator is required to place both the watermelon and the carrot into the basket. The **key points generated by GPT-4o may sometimes be missing because of hallucinations**, leading to the failure of the entire task. In **Tidy Table** task, the manipulator need to **put both the green blocks into the bowl**, which is hard for **MOKA [1]** as the inference pipeline only generates key points in the region of bbox with highest confidence to output. In contrast, based on task instructions, RoboPro can flexibly and efficiently invoke existing skills in the form of code, resulting in more stable output. For instance, RoboPro could fetch the region of different individuals of the same class through indexing on the output list by perception modules to finish **Tidy Table** task. As we have addressed in **Sec. 2 \(L115-L124\)**, there is still a gap between generated language plans and low-level robotic execution in VLM prompting method, so executable code can serve as a more expressive way to bridge high-level task descriptions and low-level execution.
>
> **Table 1. Success Rate Compared with MOKA in the Real-World Setting**
>
> |  | **Move in Direction** | **Setup Food** | **Distinct Base** | **Prepare Meal** | **Tidy Table** | **Express Words** | **Stack on Color** | **Wipe Desk** | **Average** |
> |------------|------------------------|----------------|--------------------|------------------|----------------|-------------------|---------------------|---------------|-------------|
> | **MOKA[1]** | 70.0                  | 60.0           | 40.0               | 10.0             | 0.0            | 10.0              | 10.0                | 100.0         | 37.5        |
> | **RoboPro** | 80.0                  | 90.0           | 70.0               | 60.0             | 70.0           | 60.0              | 50.0                | 100.0         | 72.5        |
>
> **Reference**
>
> [1] Fang et al., MOKA: Open-World Robotic Manipulation through Mark-Based Visual Prompting, RSS, 2024.
>
> [2] Nasiriany et al., PIVOT: Iterative Visual Prompting Elicits Actionable Knowledge for VLMs, VLMNM, 2024.

---

> ### Author Response · Authors · 2024-11-21
>
> > **Weakness 2: In the real-world experiments, there is no comparison to other baselines. Would it be possible to include a comparison against at least GPT-4o to see if RoboPro performs better?**
>
> Thanks for your suggestions. During rebuttal, we further evaluate the performance of GPT-4o with same experimental settings as RoboPro in real-world environments. As shown in **Tab. 2.**, the zero-shot success rate of RoboPro surpasses GPT-4o by 12.5 across 8 real-world tasks. On visual understanding and target identification tasks **Stack on Color** and **Tidy Table**, the performance of RoboPro significantly outperforms GPT-4o. On directional moving and one-turn tasks, GPT-4o shows comparable performance with RoboPro. The result is also consistent with our simulation experiments in **Sec. 4.1** and **Sec. 4.2**. We will supplement such results of GPT-4o on real-world environments later in our revision.
>
> **Table 2. Success Rate Compared with GPT-4o in the Real-World Setting**
>
> |   | **Move in Direction** | **Setup Food** | **Distinct Base** | **Prepare Meal** | **Tidy Table** | **Express Words** | **Stack on Color** | **Wipe Desk** | **Average** |
> |-------------|------------------------|----------------|--------------------|------------------|----------------|-------------------|---------------------|---------------|-------------|
> | **GPT-4o**  | 60.0                  | 80.0           | 80.0               | 60.0             | 40.0           | 50.0              | 10.0                | 100.0         | 60.0        |
> | **RoboPro** | 80.0                  | 90.0           | 70.0               | 60.0             | 70.0           | 60.0              | 50.0                | 100.0         | 72.5        |
>
> > **Weakness 3: While the idea of using action-free video to synthesize runtime code data is interesting, the impact of the chosen video datasets remains unclear. In this paper, the authors use DROID as their dataset—was there any specific reason for this choice? How do the type and size of the dataset affect the final performance? If I’m understanding correctly, in principle, the video data doesn’t even need to be robotic. How would the approach perform if human videos, such as Ego4D or Something-Something, were used instead? It would be interesting to include the ablation over these points.**
>
> We appreciate the reviewer's recognition of the innovation of Video2Code. Video2Code is proposed for the training of robotic policy with multimodal reasoning. The most direct video data sources for the training of RoboPro are robot-centric demonstration videos. These videos are well-aligned with the action space of robotic embodiments and primarily ensure the completeness of instruction execution. DROID is a recently proposed large-scale video demonstration dataset with abundant variance on skills, tasks and scenarios (350 hours of interaction data across 564 scenes, 86 tasks and 52 buildings), which is a direct and qualified candidate for the video data source of Video2Code. We make the first step to prove the feasibility of synthesizing runtime code data automatically using action-free videos through extensive experiments in both simulators and real-world settings.
>
> In general, human demonstrations can provide another type of potential data source for Video2Code, which definitely deserve our future investigation. However, the transition to robotic platforms is not trivial. Compared with robot-centric demonstration videos, the action space of human demonstrations are not perfectly aligned with robotic embodiments and some of demonstrations in existing human videos are incomplete. For example, Something-Something primarily contains short clips of single action, which is hard to extract procedural knowledge for efficient training of policy code generation models. We will explore this issue in the future work.
>
> > **Weakness 4: The downstream performance appears to be sensitive to the choice of VLMs used for draft/code generation, as shown in Table 6. Specifically, there is a large difference in results between using Gemini and DeepSeek for code generation. This sensitivity to the VLM choice may be a weakness of this approach.**
>
> Thanks for the comments. As stated in **Sec. 4.5 (L516-L518)**, our goal is to highlight that the enhanced visual reasoning capabilities of the Draft VLM, combined with the robust code synthesis abilities of the Code LLM, are both essential for high-quality runtime code data curation. Through the ablation studies, we aim to provide insights into designing a practical automated data curation pipeline for future works, enabling the generation of high-quality runtime code data from operational videos. Actually, our method is open to any advanced VLMs and we believe our method will find wider range of applications with the emerging development of large VLMs.

---

> > ### Author Response · Authors · 2024-11-24
> > **Further Explanation**
> >
> > **Ablation on the size of dataset:**
> >
> > For **weakness 3**, we further conducted an ablation study on the dataset size. Specifically, we trained RoboPro using 115k runtime code data collected by Video2Code from DROID, varying the dataset proportion to 25%, 50%, 75%, and 100%. We evaluated the models trained with different sizes of dataset on RLBench. As shown in **Tab.4.**, results indicate that RoboPro adheres to the scaling law: training with just 25% of the data already yields a well-performing model, while its performance continues to improve as the dataset size increases. Video2Code is efficient for scaling up of runtime code data, which deserves further exploration to involve in more robotic or processed human demonstrations.
> >
> > **Table 4. Ablation on the Size of Dataset**
> >
> > | **Training Data Proportion** | **RLBench Accuracy (%)** |
> > |------------------------------|---------------------------|
> > | 25%                         | 35.1                     |
> > | 50%                         | 36.4                     |
> > | 75%                         | 39.1                     |
> > | 100%                        | 42.7                     |

---

> ### Author Response · Authors · 2024-11-21
>
> > **Question 1: How accurate is the generated code from the data curation pipeline? Are there any errors or failure cases and how are they handled?**
>
> Thanks for your comments. Using LLMs to automatically extract information and generate data is not entirely robust. When generating plan data, we found that approximately 1% of the data exhibited repeated plan generation. These repeated plans are often excessively long and include one or more repeated steps, resulting in generated code that is redundant and verbose. To address this issue, we have implemented a rule-based filtering approach. By detecting anomalies in the length and repetitiveness of the generated plans, we identified these failed examples and regenerated the corresponding data.
>
> > **Question 2: Could the authors provide more details on the training procedure? – e.g. hyperparameters, training steps, and the computational resources and time required for training, etc.**
>
> As mentioned in **Sec. 3.3 (L278-L284)**, the training of RoboPro can be divided into three stages: visual alignment, pre-training, and supervised fine-tuning (SFT). We provide hyperparameters and training epochs in **Tab. 3.**. For the computational resources and time required for training, RoboPro is trained on 8 A100 GPUs with 80GB memory, and it takes three days in total for the training procedure. We will supplement such details in the appendix, and we will release our code and model to the public as noted in our paper.
>
> **Table 3. Details about the training procedure of RoboPro**
> | Training Stage | Global Batch Size | Learning Rate | Epochs | Max Length | Weight Decay|
> |:--------------:|:-----------------:|:-------------:|:------:|:----------:|:------:|
> |visual alignment|  256              |  1e-3         |  1     |    2048    |   0    |
> |pre-training    |  256              |  2e-5         |  1     |    2048    |   0    |
> |SFT             |  128              |  2e-5         |  1     |    3072    |   0    |
>
> > **Question 3: During the deployment phase, is it correct that only the initial image of the environment is used to generate the entire execution code, without using the newest images at each time step?**
>
> Thanks for bringing this issue to our attention. Currently, we primarily focus on manipulation tasks in the static environments, following the same settings with prior works, where using the initial image as input is sufficient for code generation. Extending to dynamic environments, such as those involving moving objects, can be effectively addressed by incorporating intermediate frames. This requires using interleaved and multi-turn image-text pairs as training data. Video2Code can also be extended to handle these types of dynamic scenarios.

---

> ### Comment · Reviewer_9oqZ · 2024-11-28
>
> I thank the authors for their response. The answers addressed most of my concerns, but I still recommend this paper as borderline accept. Therefore, I will maintain my score.

---

### Author Response · Authors · 2024-11-21
**Author Rebuttal by Authors**

**Rebuttal:**
**We sincerely thank all reviewers for the insightful feedback and for acknowledging our contributions. We have provided individual responses to your questions below. Here we'd like to highlight two points as follows.**

**1. Key contributions of this work**

- To the best of our knowledge, Video2Code is the first to leverage unstructured video contents as high-quality demonstrations for generating runtime code data in robotics.
- We developed a novel two-stage automated data pipeline that converts extensive repositories of in-the-wild operational videos into executable robotic code, substantially improving data variability and curation efficiency compared to approaches dependent on manually constructed simulation environments and human-annotated datasets.
- Attribute to the large-scale and visual-aligned runtime code data generated by Video2Code, we make an early attempt to train end-to-end policy code generation model to perceive visual information and follow free-form instructions, which is our so-called RoboPro.
- With RoboPro, we discussed zero-shot generalization ability across APIs and skills, and the impacts of visual information on vague or long-termed instructions, which is largely under-explored in prior works.

**2. Generation ability out of predefined APIs**

First, we'd like to clarify that this issue has been largely under-explored in prior policy code generation works. Training-based code generation methods, such as RoboCodeX, typically evaluate model performance using only the APIs available during the training phase.

Second, as discussed in **Sec. 4.2 \(L398-L410\)**, we have made efforts to validate the scalability of code generation methods. To the best of our knowledge, we are the first to investigate the generalization capabilities of policy code generation across different APIs and skills, extending beyond the constraints of the training-stage API set. We have examined RoboPro's robustness to variations in API structures, such as API Renaming and API Refactoring. As shown in **Tab. 1. in the main paper**, when encountering new or modified APIs during evaluation, RoboPro still demonstrates strong zero-shot adaptation capabilities. This allows it to effectively handle altered or newly defined skills without additional fine-tuning.

During Rebuttal, to further evaluate RoboPro's adaptability to newly defined or task-specific APIs, we selected three compositional tasks from RLBench that involve multi-step execution: **Water Plants**, **Hit Ball** and **Scoop Cube**. For each task, we designed a new set of task-specific APIs encompassing skills not included in RoboPro's training phase. As shown in **Tab. 1.** , the performance of RoboPro consistently outperforms GPT-4o and CaP in a zero-shot manner. This robustness is attributed to RoboPro's ability as a generalist code model to comprehend newly defined functions and sequential action knowledge learned from Video2Code. This combination enables RoboPro to seamlessly adapt to evolving API structures and new task demands, thus offering a flexible and efficient solution for robotic manipulation.

**Table 1. Success Rate Compared with SOTA Methods on Three Compositional Tasks from RLBench Based on a New Set of Task-Specific APIs**

|   | **Water Plants** | **Hit Ball** | **Scoop Cube** | **Average** |
|--------------|------------------|--------------|----------------|-------------|
| **Cap**      | 4.0             | 16.0         | 0.0            | 6.7         |
| **GPT-4o**   | 40.0            | 12.0         | 24.0           | 25.3        |
| **RoboPro**  | 40.0            | 44.0         | 48.0           | 44.0        |

---

### Author Response · Authors · 2024-11-25
**Looking forward to More Discussion**

Dear reviewers,

Thanks for your meticulous review and valuable time, which played a pivotal role in enhancing the quality of our paper. We greatly appreciate your acknowledgment of the strengths in our work,
particularly regarding the idea to use action-free data videos to synthesize robotic runtime code, comprehensive experimental evaluations. We apologize for any inconvenience, but as the deadline is approaching. We would like to provide an update on our progress.

In our rebuttal, we have addressed your concerns in detail, including issues related to generalization ability out of predefined APIs, additional experiments for other baselines (MOKA, GPT-4o, OpenVLA), differences with previous works, and ablations on the dataset size.

We hope that these clarifications resolve your confusion, and we will improve our paper on your feedback and suggestions. Please let us know if there are any additional questions or clarifications needed.

We sincerely appreciate your time and effort in reviewing our paper, and we would greatly appreciate it if you could re-consider the rating.

Best regards,

Authors.

---

### Meta-Review · Area_Chair_o4U5 · 2024-12-24

**Metareview:**

The paper introduces Robotic Programmer (RoboPro), a robotic foundation model for manipulation tasks that integrates visual perception and free-form instruction-following with policy code generation, achieving state-of-the-art zero-shot performance across simulators and real-world environments. The work also addresses data efficiency challenges through the novel Video2Code approach.

The reviewers raised several concerns about the paper, including (1) insufficient comparisons with baselines, (2) limited novelty, particularly in relation to prior work, (3) restricted applicability due to reliance on a predefined API library, (4) scalability of the method, and (5) limited analysis of the system's individual components.

During the Author-Reviewer Discussion phase, although the authors provided responses, they were unable to fully address the reviewers' concerns, leaving many issues unresolved. The AC recommends that the authors carefully revisit the reviewers' comments to improve the paper in a future revision.

**Additional Comments On Reviewer Discussion:**

In the Reviewer Discussion phase, despite Reviewer 9oqZ expressing a positive opinion of the paper, they were not willing to champion it or counter the concerns raised by other reviewers. As a result, the majority opinion remained unchanged.

---

### Decision · Program_Chairs · 2025-01-22

Reject